# DySTreSS: Dynamically Scaled Temperature in Self-Supervised Contrastive Learning

## Abstract

In contemporary self-supervised contrastive algorithms like SimCLR, MoCo, etc., the task of balancing attraction between two semantically similar samples and repulsion between two samples from different classes is primarily affected by the presence of hard negative samples. While the InfoNCE loss has been shown to impose penalties based on hardness, the temperature hyper-parameter is the key to regulating the penalties and the trade-off between uniformity and tolerance. In this work, we focus our attention on improving the performance of InfoNCE loss in Self-supervised learning by proposing a novel cosine-similarity dependent temperature scaling function to effectively optimize the distribution of the samples in the feature space. We also provide mathematical analyses to support the construction of such a dynamically scaled temperature function. Experimental evidence shows that the proposed framework outperforms the contrastive loss-based SSL algorithms. Our code is available at https://www.github.com/subanon/dystress.

## 1 Introduction

With its prowess at learning high-quality representations from large-scale unlabeled data, self-supervised learning (SSL) has revolutionized the field of machine learning. Classical approaches in SSL involved designing a suitable pretext task, such as solving jigsaw puzzles (Noroozi & Favaro, 2016), image inpainting (Pathak et al., 2016a), colorization (Zhang et al., 2016), etc. However, the problem with these methods was that there exists a significant difference between the nature of the pretext task and the desired downstream task (classification/segmentation). Recent works have focused on contrastive learning framework (He et al., 2020; Chen et al., 2020a; Yeh et al., 2022), wherein the model learns an embedding space such that the features of augmented versions of the same sample lie close to each other, pushing the embeddings of other samples farther apart. Thus, unsupervised contrastive models, aided by heavy augmentations and robust abstractions, are capable of learning certain levels of representational structures. Empirically, contrastive learning-based algorithms have been found to perform better at downstream tasks than the former SSL methods.

The most commonly used loss function for self-supervised contrastive learning (SSCL) is the InfoNCE loss, used in several works such as CPC (Oord et al., 2018), MoCo (He et al., 2020), SimCLR (Chen et al., 2020a), etc. Although the temperature hyper-parameter is an integral part of the InfoNCE function, it has mostly been trivialized as a mere scaling coefficient. Recently, in Kukleva et al. (2023) the authors have proposed a temperature hyper-parameter ($\tau$) scheduling for SSL. However, the method proposed in Kukleva et al. (2023) is mainly focused on task switching based on the temperature hyper-parameter $\tau$ rather than focusing on the effect of $\tau$ on false negative samples. In our work, however, we argue that there is more to this seemingly redundant factor. Our theoretical analyses bring forth an intuitive yet vital aspect related to the presence of constructively false negative yet inherently positive pairs – samples that do not originate from the same instance yet show a high degree of semantic representational similarity as they belong to the same underlying class. As the main objective of the contrastive loss function is to maximize the similarity of the different augmentations of the same instance while minimizing the same for different instances, the aforementioned constructively false negative pairs are repelled away. This action implies that semantic information is not an integral part of existing InfoNCE loss. Pushing the samples in semantically similar pairs away creates an adverse effect on representation learning. Large penalties on these samples along with true negative samples may increase the uniformity, but it adversely affects the alignment of the local structure constituted by samples with similar semantic information. Hence the

uniformity-tolerance (alignment) dilemma arises as addressed in Wang & Liu (2021a). In this work, we intend to dynamically scale the temperature hyper-parameter as a function of the cosine similarity to effectively control the repelling effect in these false negative pairs. We theorize that scaling the temperature dynamically will prevent disruption of the local and global structures of the feature space and improve representation learning.

The primary contributions of the proposed method called DySTreSS can be summarized as follows:

- We systematically study the role of temperature hyper-parameter and its effect on local and global structures in the feature space during optimization of the InfoNCE loss, both intuitively and theoretically, to establish the motivation for our proposed method. To the best of our knowledge, this is the first exhaustive attempt to design a temperature function that can be adaptively tuned based on local and global structures.
- With the established groundwork, we propose a temperature-scaled contrastive learning framework (DySTreSS) that dynamically modulates the temperature hyperparameter.
- We show the effectiveness of our approach by conducting experimentation across several benchmark vision datasets, with empirical results showing that our method outperforms better than several state-of-the-art SSL algorithms in the literature.

The rest of the paper is organized as follows. Sec. 2 briefs contemporary works in self-supervised learning, along with an account of the works where the temperature hyper-parameter is the focus point. Sec. 3 gives a detailed account of the theoretical background. Sec. 4 discusses the motivation and also introduces the proposed framework. Next, we present the implementation details in Sec. 5. The experimental evidence, along with various ablation studies, are presented with illustrations in Sec. 6. Finally, we conclude our work in Sec. 7.

## 2 RELATED WORK

**Self-supervised Learning.** SSL approaches (Chen et al., 2020a; Yeh et al., 2022; Jing & Tian, 2020; Bardes et al., 2022; Zbontar et al., 2021; Grill et al., 2020; Chen et al., 2020b;c; Chen & He, 2021) have become the de facto standard in unsupervised representation learning with the aim to learn powerful features from unlabelled data that can be effectively transferred to downstream tasks. Several pre-training strategies have been proposed, which can be categorized as generative or reconstruction-based (Pathak et al., 2016b; Zhang et al., 2016; Kingma & Welling, 2013; Goodfellow et al., 2014; Ledig et al., 2017), clustering-based (Caron et al., 2020; 2018; Huang et al., 2019; Bautista et al., 2016), and contrastive learning approaches (He et al., 2020; Chen et al., 2020a; Yeh et al., 2022; Chen et al., 2020b;c). Other popular methods include similarity learning (Chen & He, 2021), redundancy reduction within embeddings (Zbontar et al., 2021), and an information maximization-based algorithm (Bardes et al., 2022). Recently, self-distillation based frameworks like (Caron et al., 2021), (Zhou et al., 2022), etc. have also shown significant improvement in performance.

**Contrastive Learning.** The majority of recent SSL algorithms have leveraged contrastive learning, which causes distorted versions of the same sample to attract and different samples to repel. SimCLR (Chen et al., 2020a) simply used a contrastive loss function with a large batch size, while MoCo (He et al., 2020) leveraged momentum encoding and a dictionary-based feature bank for negative samples. These works were further enhanced in Chen et al. (2020b) and Chen et al. (2020c) respectively. Recent works like DCL (Yeh et al., 2022) where the authors decoupled the positive and negative pairing components of the InfoNCE (Gutmann & Hyvärinen, 2010) loss function.

Recently, negative-free contrastive learning frameworks like ZeroCL (Zhang et al., 2022b), WMSE (Ermolov et al., 2021), ARB (Zhang et al., 2022a) have also gained much attention. However, ZeroCL and WMSE use negative samples for calculating dimension-wise statistics. Hence, in a way, they depend on the negative samples. WMSE also involves a Cholesky decomposition step, which has a high computational complexity. The most recent work ARB improves upon the Barlow Twins method, by using the nearest orthonormal basis-based optimization objective. However, it uses a spectral decomposition step to deal with non-full rank matrices, which can be quite computationally expensive.

**Temperature in Contrastive Learning.** Recently, there have been a few pieces of work that have focused on the temperature hyper-parameter in the InfoNCE loss function. In Zhang et al. (2021),

the authors present a temperature hyper-parameter as a function of the input representations thereby incorporating uncertainty in the form of temperature. Wang & Liu (2021a) explores the hardness-aware property of contrastive loss and the role of temperature in it by measuring the uniformity and tolerance of representations. On the other hand, MACL (Zizheng et al., 2023) assumed the temperature hyperparameter as the function of alignment to address the uniformity-tolerance dilemma that exists in the InfoNCE loss design. Motivated by the study shown in Wang & Liu (2021a), the authors in Kukleva et al. (2023) proposed a continuous task switching between instance discrimination and group-wise discrimination by using simple cosine scheduling. In Qiu et al. (2023), the authors attempt to implement distributionally robust optimization for individual temperature individualization, that is, it uses a temperature hyper-parameter $\tau_i$ corresponding to each anchor sample $x_i$, and is updated at each iteration. This no longer keeps the temperature as a hyper-parameter and converts it into another optimizable parameter.

## 3  THEORETICAL BACKGROUND

In self-supervised contrastive learning (Oord et al., 2018; Chen et al., 2020a; He et al., 2020), the InfoNCE loss is given by Eqn. 1.

$$\mathcal{L} = \sum_i \mathcal{L}_i = -\sum_i ln\left(p_{ii+}\right) = -\sum_i ln\left(\frac{exp(\frac{s_{ii+}}{\tau})}{\sum_j exp(\frac{s_{ij}}{\tau})}\right) \tag{1}$$

where $ii+$ denotes a true positive pair and $s_{ij}$ is the cosine similarity between the latent vectors of the samples $x_i$ and $x_j$. In self-supervised contrastive learning frameworks like SimCLR (Chen et al., 2020a), MoCo (He et al., 2020), etc., we assume that each sample is a class on its own. This results in the pairing of any two samples that may belong to the same class, resulting in the formation of false negative (FN) pairs. The similarity between these samples in these types of pairs can assume high cosine similarity values. On the contrary, pairs consisting of two samples belonging to two different classes comprise true negative (TN) pairs. However, depending on the mapping of the corresponding features to the feature space, true negative pairs can also have high cosine similarity between the constituent samples, and are called hard true negative pairs. False negative pairs by construction can also act as hard false negative pairs. True positive (TP) pairs are simply constituted of samples obtained by two random augmentations of a sample in the dataset.

## 4  METHODOLOGY

### 4.1  ROLE OF TEMPERATURE IN CONTRASTIVE LEARNING

InfoNCE loss concentrates on optimization by penalizing the hard negative pairs according to their hardness (Wang & Liu, 2021a). The gradient of $\mathcal{L}_i$ w.r.t. $s_{ii}$ and $s_{ij}$ is given by Eqn. 2 and 3. A simple relative weightage is defined in Wang & Liu (2021a) and as given in Eqn. 4.

$$\frac{\partial \mathcal{L}_i}{\partial s_{ii+}} = -\frac{1}{\tau}\sum_{k\neq i} p_{ik} \quad (2) \qquad \frac{\partial \mathcal{L}_i}{\partial s_{ij}} = \frac{1}{\tau}p_{ij} \quad (3) \qquad r(s_{ij}) = \frac{|\frac{\partial \mathcal{L}_i}{\partial s_{ij}}|}{|\frac{\partial \mathcal{L}_i}{\partial s_{ii+}}|} = \frac{exp(\frac{s_{ij}}{\tau})}{\sum_{k\neq i} exp(\frac{s_{ik}}{\tau})} \quad (4)$$

Therefore, the role of temperature $\tau$ in contrastive loss is to control the relative weightage of the hard negative samples. As a result, low temperature values tend to penalize more without semantic similarity awareness and create a more uniformly distributed feature space. Hence, it is evident that the temperature hyper-parameter acts as the control knob for the uniformity.

### 4.2  EFFECT OF TEMPERATURE ON LOCAL AND GLOBAL STRUCTURES

Let us assume that for a given sample $x$, we have an encoder $f: x \rightarrow z \in \mathbb{R}^D$, where $z$ is a mapped image. Under any valid distance measure $\mathfrak{M}$ on the manifold $\mathfrak{U}$ of $z$, in an optimal scenario, if convergence is achieved in a self-supervised pre-training stage, two mapped images $z_i$ and $z_j$, where $i \neq j$, from same class $C$, will have minimum possible distance. In our work, the term 'local structure' of any sample refers to the arrangement of the other samples in the close local neighborhood of that

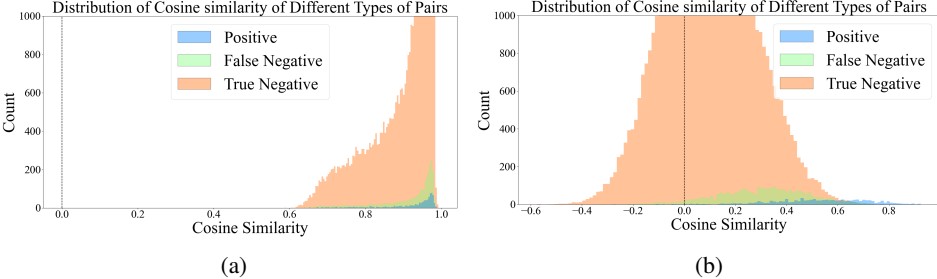

Figure 1: (a) Histogram of cosine similarities of true positive (TP), false negative (FN), and true negative (TN) pairs at random initialization, (b) Histogram of cosine similarities of TP, FN, and TN pairs after pre-training. (Best viewed at 300%)

sample and can be denoted by the samples included in a closed ball of radius $r_j$ around centroid sample $x_j$. Likewise, the term 'global structure' takes the arrangement of all the samples in the feature space into account.

As already stated in Sec. 4.1, decreasing the temperature tends to penalize the hard negative pairs more. This is because hard negative pairs tend to have high cosine similarity (say, $s_{ij}$). With small temperature, the quantity $\frac{s_{ij}}{\tau}$ is further amplified, consequently resulting in a larger penalty (from Eqn. 4). This causes samples constituting false negative pairs to drift apart. Consequently, the local structure consisting of samples of any particular class is disturbed. This effect on the false negative pairs gives rise to the "uniformity-tolerance dilemma". From Fig. 1a, we can see the distribution of the cosine similarity of the true negative (TN) and false negative (FN) pairs at the start of pre-training, and we can observe that there is little difference between the mode of the true and false negative pair histogram, making the task difficult. The distribution of cosine similarity at the end of pre-training is shown in Fig. 1b where we can see the shift in the distribution of cosine similarity of true negative pairs. Empirically, the ratio of the earth mover's distance (EMD) between TN and FN pairs, and TP and FN pairs is greater than 1, whereas the ratio of EMD between TN and FN pairs, and TP and FN pairs is less than 1, which indicates that the TN pairs are moved farther apart than FN pairs.

The effect of temperature can be better understood if we take the gradient of the loss with respect to any latent vector $z_j$. Taking the expression of the derivative of the loss $\mathcal{L}$, given by Eqn. 1, with respect to $z_j$, we get (See App. Sec. F for the intermediate step),

$$\frac{\partial \mathcal{L}}{\partial z_i} = - \left[ \frac{z_{i+}}{\tau} \left( 1 - p^{ii^+} \right) - \sum_{\substack{j=1 \\ j \neq i}}^{N} \frac{z_j}{\tau} \left( p^{i \Downarrow j} + p^{j \Downarrow i} \right) \right] \tag{5}$$

where $C_{ij} = \frac{s_{ij}}{\tau}$, denotes the cosine similarity between the feature vectors $z_i$ and $z_j$, scaled by temperature $\tau$, and $(z_i, z_{i+})$ forms the positive pair. The quantity $p^{i \Downarrow j}$ is the probability of the pair $(x_i, x_j)$ being predicted as a positive pair with the sample $x_i$ as the anchor. Hence, from the expression of the displacement vector $\frac{\partial \mathcal{L}}{\partial z_i}$, we can arrive at the conclusion that at a low-temperature value the sample $z_i$ moves away from any sample $z_j$ if they are mapped close to each other in the feature space. In other words, contrastive loss penalizes hard negative pairs. The effect of temperature reduction in different scenarios is discussed as follows.

**Reducing Temperature for False Negatives:** By Gradient Descent rule, $z_i^{t+1} = z_i^t - \frac{\partial \mathcal{L}}{\partial z_i^t}$. The contribution of false negative pairs should be negative to the gradient of $\mathcal{L}$ with respect to the feature (latent) vector $z_i$. For false negative pairs, the value of cosine similarity between the two elements in the pair can be positive or negative depending on where the samples are mapped. Adjusting the temperature hyper-parameter allows us to control the contribution of the false negative pairs in the loss optimization process, by scaling the weights of the latent vectors. For two closely placed false negative samples, the sum of the corresponding probabilities $p^{i \Downarrow j} + p^{j \Downarrow i}$ as shown in Eqn. 5 will be high. If the temperature is decreased, the contribution of the sample $z_j$ in the gradient increases further, resulting in the sample $z_i$ drifting opposite to the direction of $z_j$. Conversely, if we take the

derivative of $\mathcal{L}$ with respect to $z_j$ similar to Eq. 5, we will get a term involving $z_i$, which will enforce a similar effect on $z_j$. This results in the disruption of the local cluster structure in the feature space.

**Reducing Temperature for Hard Negatives:** For hard negative pairs, we can expect the two constituent samples to drift apart from each other if a low enough temperature is applied. However, without a ground truth label, it is impossible to apply selective temperature moderation to all the pairs. If we decrease the temperature for all pairs whose cosine similarity is above a certain threshold (say, $C_\alpha$), then the closely spaced false negative pairs will also be affected, resulting again in disruption of the local cluster structure.

**What if there were no False Negatives?** In an ideal scenario, where all the negative pairs are true negative pairs (like in supervised contrastive learning (Khosla et al., 2020)), we may make the mistake of assuming that we can safely decrease the temperature. Decreasing the temperature for true negative pairs will certainly improve performance up to a certain level, below which the performance degrades due to numerical instability (Khosla et al., 2020), as the gradients become too large. This degradation in performance is due to the disruption in the global structure of the feature space. Disruption in local structure causes degradation of alignment in the feature space, whereas disruption in the global structure will cause an increase in uniformity (Wang & Liu, 2021a; Wang & Isola, 2020).

**Increasing Global Temperature:** On the other hand, increasing the temperature for all the samples has the opposite effect. As the temperature is increased, the drift in the false negative pairs is reduced, thereby helping in maintaining proper alignment. However, the uniformity may be affected as the repulsion between samples constituting true negative pairs including the hard true negatives, will also be reduced. Hence, increasing temperature causes an increase in alignment but affects uniformity.

### 4.3 MOTIVATION OF PROPOSED TEMPERATURE SCALING FUNCTION

In SSCL, we can neither know for certain the boundary between true and false negatives, nor the class labels. However, we have described the effect of temperature on the feature space in the Sec. 4.2 and can list some criteria we need to follow to design a proper temperature scaling function. The criteria are as follows: (1) A very low temperature in the highly negative cosine similarity region will disrupt the global structure, (2) A very low temperature in the highly positive cosine similarity region will disrupt the local structure, (3) A very low temperature for false negative pairs, which we can assume to lie in the range $[s_{fn}, +1.0]$ can affect hard true negatives and true positives pairs, where $s_{fn}$ denotes a cosine similarity score, (4) A high temperature will affect the uniformity of the feature space and delay convergence.

Let us assume $\tau(\cdot)$ is the temperature function, which takes the cosine similarity of a pair as input and outputs a temperature value for the same. For the rest of this literature, we will consider $\tau_{ij} = \tau(s_{ij})$.

$$\frac{\partial \mathcal{L}_i}{\partial s_{ii+}} = -\frac{\tau_{ii} - s_{ii+}\frac{\partial \tau_{ii}}{\partial s_{ii+}}}{\tau_{ii}^2} \cdot (1 - p_{ii+}) \quad (6) \qquad \frac{\partial \mathcal{L}_i}{\partial s_{ij}} = \frac{\tau_{ij} - s_{ij}\frac{\partial \tau_{ij}}{\partial s_{ij}}}{\tau_{ij}^2} \cdot p_{ij} \quad (7)$$

where $s_{ii+}$ and $s_{ij}$ denote the cosine similarity of positive and negative pairs, respectively.

For negative pairs, $\frac{\partial \mathcal{L}}{\partial s_{ij}} = \delta > 0$, where $\delta$ is a non-negative number. From Eqn. 7, we get,

$$\frac{\partial \mathcal{L}}{\partial s_{ij}} = \frac{\tau_{ij} - s_{ij}\frac{\partial \tau_{ij}}{\partial s_{ij}}}{\tau_{ij}^2} p_{ij} = \delta \quad \text{where } \delta < \epsilon \text{ and } \delta, \epsilon > 0 \quad (8)$$

Now, we will try to build a few assumptions about our temperature function, and later we will show that these assumptions hold true, by solving a differential equation arising from Eqn. 6 and 7. Without loss of generality, we can always assume $\tau_{ij} > 0$. As the temperature parameter cannot be negative, the temperature value would be 0.0 or some positive constant. For $s_{ij} < 0$, to satisfy our criteria (1) and (3), we should have $\frac{\partial \tau_{ij}}{\partial s_{ij}}$ is always less than some negative number. Hence, the slope of the temperature function is negative in the negative half of the cosine similarity vs. temperature plane. In the positive half of the cosine similarity vs. temperature plane, the slope of the temperature function is less than some positive number. However, a negative slope in the positive half would mean that temperature would decrease at high cosine similarity, violating our criteria (2) and (3). A low temperature at high cosine similarity will affect the hard negative pairs and degrade the local

structure. Therefore, we should adopt the temperature function, such that the temperature does not violate criteria (4) at high cosine similarity values.

**Proposition 1:** *The temperature function should have a negative and positive slope in the negative and positive half of the cosine similarity vs. temperature plane, respectively.*

**Proof:** *The proof is given in detail in Appendix C.*

As shown in Fig. 8 in Appendix B, we can observe that the plotted curves show positive and negative gradients on the positive and negative half of the cosine similarity vs. temperature plane, respectively, as stated in this section. This establishes our intuitively derived assumptions for the slopes of the temperature function in the negative and positive halves of the cosine similarity vs. temperature plane. The detailed proof of the proposition is given in Appendix C.

### 4.4 PROPOSED FRAMEWORK

Combining all the above philosophies together we describe the framework proposed in this work. In this work, we use SimCLR (Chen et al., 2020a) as the baseline framework. To satisfy the conditions derived in the section above, we adopt a cosine function of the cosine similarity as the temperature function, as shown in Alg. 1. We can visualize from Fig. 2, that the cosine function does not violate the four criteria mentioned in Sec. 4.3.

---

**Algorithm 1:** Temperature Scaling Function

**Data:** $\tau_{max}$ and $\tau_{min}$
**Input:** $s_{ij} \to$ Cosine Similarity of the pair $(x_i, x_j)$
$\tau_{ij} = \tau_{min} + 0.5 \times (\tau_{max} - \tau_{min}) \times (1 + \cos(\pi(1 + s_{ij})))$

---

For different $\tau_{max}$ and $\tau_{min}$ values, we obtain different temperature scaling functions as shown in Fig. 2. We will analyze how a different temperature scaling scheme affects performance in Sec. A.3.

Assigning lower temperatures to FN pairs, pushes the constituent samples far apart. To reduce this effect, we can shift the minimum of the temperature function into the negative half of the cosine similarity vs. temperature plane. The result of this modification can be seen in Table 7. As the contribution corresponding to the FN pairs reduces in $\frac{\partial \mathcal{L}}{\partial z_i}$ in Eqn. 5, the performance also improves.

## 5 IMPLEMENTATION DETAILS

**Datasets** To study the effects of temperature in the self-supervised contrastive learning framework, we used the ImageNet1K (Deng et al., 2009) and ImageNet100 (Tian et al., 2020) datasets. We also used 3 small-scale datasets, namely, CIFAR10 (Krizhevsky, 2009) and CIFAR100 (Krizhevsky, 2009).Furthermore, we also study the effect of our proposed framework on the Long-tailed versions of the aforementioned datasets, which we term CIFAR10-LT and CIFAR100-LT (Appendix A.2).

**Pre-training Details** For the experiments on ImageNet1K and ImageNet100 datasets, we used a ResNet50 (He et al., 2016) backbone for all our experiments. We optimized the network parameters using a LARS optimizer with the square root learning rate scaling scheme as described in the SimCLR (Chen et al., 2020a) paper. For all our experiments we used a batch size of 256, The pre-training and the downstream tasks were run on a single 24GB NVIDIA A5000 GPU using the lightly-ai (Susmelj et al., 2020) library. To ensure faster training and prevent out-of-memory issues, we adopted automatic mixed precision (AMP) training. The pre-training details for the CIFAR datasets are provided in the appendix.

**Augmentations** During the pre-training stage, two augmented versions of each input image are generated and used as positive pairs. For the augmentations, we follow the augmentation strategy of SimCLR (Chen et al., 2020a).

# 6 EXPERIMENTAL RESULTS AND ABLATIONS

We conducted extensive experiments on the ImageNet100, ImageNet1K, CIFAR10, and CIFAR100 datasets to empirically prove that the proposed temperature scaling framework outperforms the state-of-the-art SSL frameworks and also the recent temperature modulating framework MACL (Zizheng et al., 2023). For pretraining on sentence embeddings, we used the Wiki1M dataset (Gao et al., 2021).

## 6.1 RESULTS ON IMAGENET100

We consider the vanilla SimCLR as the baseline for our proposed temperature scaling framework, hence all the experiments on ImageNet100 were conducted on the SimCLR framework. We also compare our work with the state-of-the-art contrastive learning frameworks. In Tab. 1, we present results of the vanilla DySTreSS framework and a shifted version of the same (Sec. 6.5.2) with shift and scale parameters $\triangle s$ and $k$, respectively. The values of $\tau_{min}$ and $\tau_{max}$ were set to 0.1 and 0.2, respectively. ✓ and ✗ in the "Temp. Scaled" column indicates if the corresponding framework uses temperature scaling. SimCLR was used as the baseline in DCL, MACL, and DySTreSS.

Table 1: Comparison with state-of-the-art SSL frameworks on ImageNet100 dataset. For DySTreSS*, $\triangle s = -0.4, k = 0.7$. (Here, B. Twins stands for Barlow Twins)

| Framework | Temp. Scaled | Lin. Eval. Acc. | |
|---|---|---|---|
| | | Top-1 | Top-5 |
| SimCLR | ✗ | 75.54 | 93.06 |
| DCL | ✗ | 77.38 | 94.01 |
| BYOL | N/A | 75.02 | 93.42 |
| B. Twins | N/A | 75.88 | 93.96 |
| VicReg | N/A | 76.36 | 94.14 |
| MACL | ✓ | 78.28 | 94.25 |
| DySTreSS | ✓ | 78.78 | 94.70 |
| DySTreSS* | ✓ | **78.82** | **94.76** |

Table 2: Comparison with state-of-the-art SSL frameworks on ImageNet1K dataset (Here, B. Twins stands for Barlow Twins)

| Framework | Temp. Scaled | Lin. Eval. Acc. | |
|---|---|---|---|
| | | Top-1 | Top-5 |
| SimCLR | ✗ | 63.2 | 85.2 |
| DCL | ✗ | 65.1 | 86.2 |
| DCLW | ✗ | 64.2 | 86.0 |
| BYOL | N/A | 62.4 | 82.7 |
| B. Twins | N/A | 62.9 | 84.3 |
| VicReg | N/A | 63.0 | 85.4 |
| MACL | ✓ | 64.3 | - |
| DySTreSS | ✓ | **65.21** | **86.55** |

We observe that the proposed framework outperforms the contemporary state-of-the-art SSL methods on linear probing evaluation. The proposed framework also outperforms the recent state-of-the-art framework MACL (Zizheng et al., 2023) which also adopts a temperature-modifying approach on top of the contrastive learning SimCLR framework. Furthermore, we also applied the proposed DySTreSS framework on the SimCLR+DCL framework with the base configuration $\tau_{min} = 0.1$ and $\tau_{max} = 0.2$, and achieved an improvement of $0.14\%$ over the Top-1 accuracy reported in Table 1.

## 6.2 RESULTS ON IMAGENET1K

As the implementation is done using the lightly-ai library, we use the benchmark results provided by the library on the ImageNet1K dataset for the comparison. The results shown in Table 2 are for 100 epochs of pre-training on IamgeNet1K. We observe that the proposed framework outperforms the contemporary state-of-the-art self-supervised methods on the ImageNet1K dataset with 100 epochs of pre-training on linear probing accuracy. The results in Table 2 indicate that pre-training with the proposed framework provides better representations that can be easily separable by a linear classifier.

## 6.3 RESULTS ON SMALL SCALE BENCHMARKS

In this section, we present the performance of our proposed framework on CIFAR10 and CIFAR100(Tab. 3) datasets. We have reported the best results obtained with $\tau_{min} = 0.07$ and $\tau_{max} = 0.2$ for comparison in Tab. 3. From our ablations on CIFAR datasets (App. A.3), we observed that the vanilla DySTreSS works better than the shifted versions. On the long-tailed datasets (Tab. 4), the DySTreSS outperforms Kukleva et al. (2023) after 2000 epochs of pre-training.

On the CIFAR-10 dataset, the proposed framework outperforms the recent state-of-the-art (SOTA) framework MACL (Zizheng et al., 2023) along with other SSL contrastive frameworks. On the CIFAR-100 dataset, DySTreSS outperforms SimCLR and MACL by 1.77% and 1.39%, respectively.

Table 3: Comparison with SOTA SSL frameworks on CIFAR10 and CIFAR100 datasets.

| Framework | Temp. Scaled | CIFAR10 | CIFAR100 |
|---|---|---|---|
| SimCLR | ✗ | 83.65 | 52.32 |
| MoCoV2 | ✗ | 83.9 | 54.01 |
| SimCLR+DCL | ✗ | 84.4 | 56.02 |
| MACL (repro.) | ✓ | 84.85 | 56.15 |
| DySTreSS | ✓ | **85.68** | **56.57** |

Table 4: Comparison with SOTA SSL frameworks on long-tailed datasets.

| Framework | CIFAR10-LT | CIFAR100-LT |
|---|---|---|
| Kukleva et al. (2023) | 62.91 | 30.20 |
| DySTreSS | **64.98** | **31.71** |
| 200 epochs pre-training | | |
| SimCLR | 55.29 | 26.18 |
| DySTreSS | **56.40** | **27.10** |

## 6.4 EXPERIMENTS ON SENTENCE EMBEDDING

Similar to MACL (Zizheng et al., 2023), we adopt SimCSE (Gao et al., 2021) as the baseline for sentence embedding learning. We conduct experiments with BERT (Devlin et al., 2019) on Semantic Textual Similarity (STS) and Transfer tasks following Gao et al. (2021). We can observe from Table 5 that the proposed framework achieves better performance on most STS and transfer tasks over the SimCSE and MACL baseline under the same experimental environment. In support of the difference in reproduced results from the reported results in the respective papers, we speculate the cause to be the difference in the hardware, as already mentioned in Zizheng et al. (2023). All the experiments were conducted on a 24GB NVIDIA A5500 GPU.

Table 5: Comparison on STS and Transfer tasks (metric: Spearman's correlation with "all" setting). DySTreSS* means DySTreSS with $\triangle s = -0.4, k = 0.7, \tau_{min} = 0.03$ and $\tau_{max} = 0.05$

| STS Task | STS12 | STS13 | STS14 | STS15 | STS16 | STSB | SICKR | Avg. |
|---|---|---|---|---|---|---|---|---|
| SimCSE (repro.) | 65.10 | 80.32 | 71.42 | 80.51 | **77.80** | 76.47 | 70.69 | 74.62 |
| w/ MACL (repro.) | 67.49 | 81.55 | 73.21 | 80.97 | 77.52 | 76.54 | 70.87 | 74.84 |
| w/ DySTreSS | 68.00 | 81.58 | 73.44 | 80.33 | 77.65 | 76.24 | 70.98 | 74.80 |
| w/ DySTreSS* | **69.34** | **81.76** | **73.51** | **81.61** | 77.39 | **76.67** | **71.53** | **75.96** |
| Transfer Task | MR | CR | SUBJ | MPQA | SST2 | TREC | MRPC | Avg. |
| SimCSE (repro.) | 80.42 | 85.80 | 94.13 | 88.66 | 85.17 | 87.40 | 72.29 | 84.84 |
| w/ MACL (repro.) | 80.50 | 85.41 | 94.20 | 89.22 | 85.06 | 90.40 | **75.59** | 85.77 |
| w/ DySTreSS | 81.30 | 87.10 | 94.79 | 89.15 | **86.16** | 88.20 | 72.35 | 85.58 |
| w/ DySTreSS* | **81.69** | **87.28** | **94.80** | **89.29** | 85.83 | **91.20** | 72.93 | **86.15** |

## 6.5 ABLATION STUDIES

### 6.5.1 EFFECT OF TEMPERATURE RANGE

To find the optimal range of temperature, we conduct pre-training with several temperature ranges on the ImageNet100 dataset as given in Table 6 and find that the values $\tau_{min} = 0.1$ and $\tau_{max} = 0.2$ gives the best performance. We observe that setting the value of $\tau_{max}$ towards high degrades performance, whereas setting both $\tau_{max}$ and $\tau_{min}$ towards low also degrades performance. The reason for such behavior can be understood from the uniformity and tolerance plots in Fig. 3. For the settings given in Tab. 6, the interclass uniformity (See Eqn. 19, App. Sec. E) are -0.6584, -0.7202, -0.7422, -0.7710, respectively. As theorized, for the worst-performing case, we can see that due to low temperature, the tolerance is high but the interclass uniformity is high too, deviating from the ideal global structure (Sec. 4.2). Whereas for the second worst-performing case, we can see that due to the high temperature, the global structure is again disrupted as an increase in uniformity means the samples are more spread out, gain in accuracy is solely due to a decrease in interclass uniformity. In the rest of the cases, tolerance is high and the interclass uniformity is also low, indicating that the interclass distance is high, facilitating better classification.

### 6.5.2 EFFECT OF SHIFTED TEMPERATURE PROFILES

Similar to the previous ablation study, we look to find the optimal temperature profile over the cosine similarity spectrum by shifting the minima of the temperature profile to the left or right as depicted in Fig. 4. When applying a shift ($\triangle s$) to the minima at $s = 0.0$, we also scale the temperature profile by $k$ to ensure that the temperatures at the extremities of the cosine similarity spectrum ($s = \pm 1.0$) remain at $\tau_{max}$ according to our assumptions in Sec. 4.2.

Table 6: Ablation of DySTreSS on different temperature ranges on ImageNet100 dataset.

| $\tau_{min}$ | $\tau_{max}$ | 20NN Acc. | | Lin. Eval. Acc. | |
|---|---|---|---|---|---|
| | | Top-1 | Top-5 | Top-1 | Top-5 |
| 0.07 | 0.1 | 67.18 | 87.82 | 77.28 | 94.16 |
| 0.07 | 0.5 | 67.88 | 88.22 | 76.34 | 93.72 |
| 0.07 | 0.2 | 71.46 | 89.42 | 78.46 | 94.4 |
| 0.1 | 0.2 | 72.00 | 89.76 | 78.76 | 94.7 |

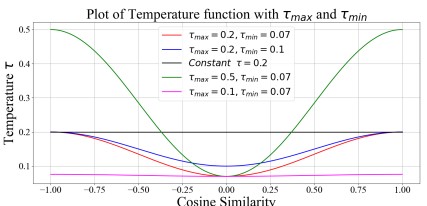

Figure 2: Temperature functions for different $\tau_{max}$ and $\tau_{min}$. Best visible at 200%.

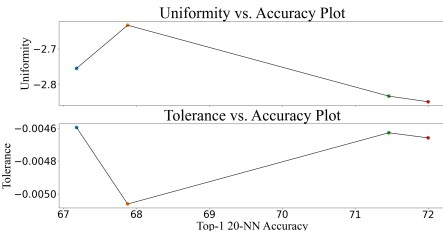

Figure 3: Plot of Uniformity and Tolerance vs. 20NN Top-1 accuracy shown in Tab. 6 The figure is best visible at 300%.

---

**Algorithm 2:** Shifted Temperature Functions

**Data:** $\tau_{max}, \tau_{min}, \Delta\tau = \tau_{max} - \tau_{min}, \Delta s, k$
**Input:** $s_{ij} \to$ Cosine Similarity of $(x_i, x_j)$
**if**
$(\Delta s \leq 0 \wedge s_{ij} \leq -\Delta s) \vee (\Delta s \geq 0 \wedge s_{ij} \geq -\Delta s)$
**then**
$\quad \tau_{ij} = \tau_{min} + \frac{\Delta\tau}{2}(1 + \cos(\frac{\pi}{k}(\Delta s + s_{ij})))$
**else**
$\quad \tau_{ij} = \tau_{max}$

---

We primarily use three different shifted versions of the temperature profile for the ablations on the ImageNet100 dataset. In the first version, Shifted Minima Ver. 1, we shift the minima towards the right half plane. In the second version, Shifted Minima Ver. 2, we shift the minima towards the left half plane. In the last and third versions, we shift the temperature profile entirely in the left half plane and keep the temperature constant in the right half plane. In addition to the shift, we apply appropriate scaling such that the extremities have the maximum temperature. The algorithm for calculating the temperature for the shifted temperature profile is given in Alg. 2. We observe that a shift of $\triangle s = -0.4$ from $s = 0.0$ and a scaling of $k = 0.7$ yields the best linear evaluation performance on the ImageNet100 dataset.

Table 7: Ablation on different temperature profiles on ImageNet100 dataset.

| shift | scale | 20-NN Accuracy | | Lin. Eval. Acc. | |
|---|---|---|---|---|---|
| | | Top-1 | Top-5 | Top-1 | Top-5 |
| 0.0 | 0.5 | 71.66 | 90.16 | 78.46 | 94.86 |
| 0.2 | 0.6 | 71.58 | 89.14 | 78.56 | 94.38 |
| 0.4 | 0.7 | 71.58 | 89.58 | 78.46 | 94.54 |
| -0.2 | 0.6 | 72.38 | 89.99 | 78.78 | 94.59 |
| -0.4 | 0.7 | 71.58 | 90.16 | 78.82 | 94.76 |

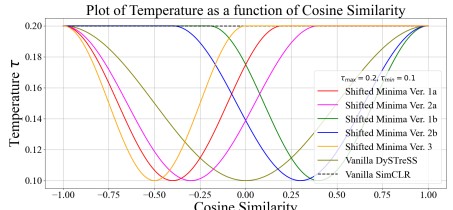

Figure 4: Plot of shifted versions of Temperature functions. Best visible at 200%.

## 7 CONCLUSION

In this work, we identified a specific category of pairs in self-supervised contrastive learning and analyzed the effect of temperature on such pairs in the optimization of the InfoNCE loss. We observed that by varying the temperature as a function of the cosine similarity values of the feature vectors of all pairs, we can control the dynamics of the optimization process and improve the performance of the baseline method, SimCLR. Through extensive experiments, we show that the proposed framework improves performance over the baseline and state-of-the-art algorithms. Finally, this work lays the foundation for further research into the working principle and dynamics of the InfoNCE loss function.

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

# A   SMALL SCALE BENCHMARKS

## A.1   TRAINING DETAILS

The encoder used in the pre-training model for experiments on CIFAR10 and CIFAR100 is ResNet18 (He et al., 2016). The first convolutional layer in ResNet18 is replaced by a convolutional layer with a kernel of dimension $3 \times 3$ and the subsequent Max-pooling layer is removed. Similarly, for CIFAR10-LT and CIFAR100-LT, we used the aforementioned ResNet18, as well. However, for the experiments on Tiny-ImageNet, we use the original ResNet50 (He et al., 2016). The last fully connected layer from the ResNet network is removed for all experiments, and the output obtained from the ResNet encoder is fed into a 2-layer multi-layered perceptron (MLP) network called Projector. For the projector architecture, we follow the SimCLR (Chen et al., 2020a), where the Linear layers are followed by Batch Normalization (BN) (Ioffe & Szegedy, 2015) layers, with a ReLU (He et al., 2015) activation function in between the first BN layer and the second Linear layer. For the pre-training procedure, we use SGD optimizer (momentum= $0.9$, weight decay factor= $5e - 4$) with a learning rate of $0.06$ and batch size of 128 for all the datasets. For the balanced CIFAR and Tiny-ImageNet datasets, we run the optimization procedure for 200 epochs. For the Long-tailed versions of the CIFAR datasets, we adopt 500 epochs of training (Kukleva et al., 2023).

For the evaluation stage, we adopt a kNN classifier. For the balanced CIFAR and Tiny-ImageNet datasets, we used a kNN classifier with $k = 200$, with cosine similarity-based weights. For the Long-tailed versions of CIFAR datasets, we used a k value of 1 and 10 with $L2$-distance-based weights. All the training and inference were run on a 16GB NVIDIA P100 GPU. Since the proposed framework is based on InfoNCE, the computation overhead is the same as contemporary frameworks such as SimCLR (Chen et al., 2020a), MoCov2 (Chen et al., 2020c), etc.

## A.2   EXPERIMENTAL RESULTS ON LONG-TAILED DATASETS

On the long-tailed versions of the CIFAR datasets, the proposed framework improves upon the baseline SimCLR by more than 1%, as seen in tables 8 and 9. ✓ and ✗ in the "Temp. Scaled" column indicates if the corresponding framework uses temperature scaling. We also see in Table 10, that our proposed method performs better than Kukleva et al. (2023) when pre-trained for 2000 epochs, as per the pre-training configuration in Kukleva et al. (2023). For the experiments on ImageNet100-LT, we used a batch size of 256 for both Kukleva et al. (2023) and DySTreSS.

Table 8: Comparison with state-of-the-art SSL frameworks on CIFAR10-**LT** dataset. DySTreSS and DySTreSS* both have $\tau_{max} = 0.2$, but the values of $\tau_{min}$ are 0.07 and 0.1, respectively.

| Framework | Temp. Scaled | Accuracy | |
|---|---|---|---|
| | | 1-NN | 10-NN |
| SimCLR | ✗ | 57.12 | 55.29 |
| DySTreSS | ✓ | **58.36** | 56.40 |
| DySTreSS* | ✓ | 58.34 | **56.54** |

Table 9: Comparison with state-of-the-art SSL frameworks on CIFAR100-**LT** dataset. DySTreSS and DySTreSS* both have $\tau_{max} = 0.2$, but the values of $\tau_{min}$ are 0.07 and 0.1, respectively.

| Framework | Temp. Scaled | Accuracy | |
|---|---|---|---|
| | | 1-NN | 10-NN |
| SimCLR | ✗ | 28.27 | 26.18 |
| DySTreSS | ✓ | **29.43** | 27.10 |
| DySTreSS* | ✓ | 28.82 | **27.32** |

Table 10: Comparison with Kukleva et al. (2023) on benchmark datasets

| Framework | CIFAR10-LT | CIFAR100-LT | ImageNet100-LT |
|---|---|---|---|
| Kukleva et al. (2023) | 62.91 | 30.20 | 45.3 (repro.) |
| DySTreSS | **64.98** | **31.71** | **46.1** |

### A.3 Ablation Studies on CIFAR datasets

#### A.3.1 Effect of Different Temperature Ranges

The performance of the proposed framework also varies as the values of $\tau_{max}$ and $\tau_{min}$ are varied. We experimented with different temperature ranges as given in Fig. 5 and observed that the proposed framework achieves the best accuracy in the temperature range $\tau_{min} = 0.07$ to $\tau_{max} = 0.2$ for both the CIFAR datasets. The different temperature range affects different types of samples differently. It is evident from Fig. 5, an increase in temperature causes a decrease in uniformity. For example, for temperature ranges($(\tau_{min}, \tau_{max})$), $(0.2, 0.5)$ and $(0.2, 0.2)$, we see a decrease in uniformity and an increase in tolerance. Whereas, for temperature ranges with lower $\tau_{min}$ but the same $\tau_{max}$, the general trend shows that the uniformity is greater and tolerance is lower. This conforms with our theory in Sec. 4.2. On the contrary, for CIFAR100, due to the presence of *more true negative pairs* than CIFAR10, increasing temperature inhibits the uniformity (tolerance) from increasing (decreasing) sufficiently to improve performance. Hence, for the same $\tau_{min}$, the performance was better for a lower $\tau_{max}$.

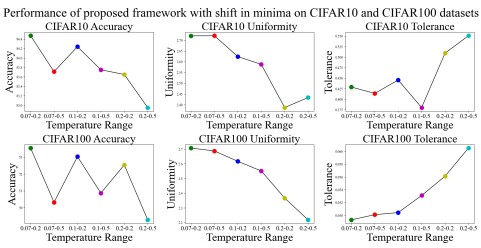

Figure 5: Plot of Accuracy on CIFAR10 (top) and CIFAR100 (bottom) datasets for different temperature ranges. The figure is best visible at 500%.

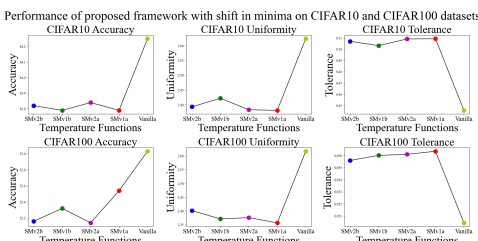

Figure 6: Plot of Accuracy, Uniformity, and Tolerance with a shift in minima for CIFAR10 (top) and CIFAR100 (bottom) dataset. The colour codes are matched to the curves in Fig. 4. The figure is best visible at 500%.

#### A.3.2 Effect of Shift in the Minimum

In Fig. 6, we present the accuracy, uniformity, and tolerance values for different temperature functions given in Fig. 4. We observe that shifting the minimum of the temperature function influences different samples and consequently changes the structure of the feature space and performance accordingly. Along the x-axis in Fig. 6, 'SMvx' denotes 'Shifted Minima Version x'. 'Ver. xa' and 'Ver. xb' denotes two shifts of $-0.2$ and $-0.4$ from the origin. For the CIFAR10 dataset, we can observe that a shift of $-0.2$ is better than a shift of $-0.4$, while the reverse is true for the CIFAR100 dataset. However, none of the configurations yields better results than the vanilla version with no shift. A lower temperature towards $s_{ij} = -1$ increases uniformity, as evident from the difference in uniformity between SMv1b and SMv2b or SMV1a and SMv2a, while the reverse is true for tolerance. The drop in performance is primarily due to the fact that a constant temperature in the range $[-\tau_{shift}, 1.0] \mid (\tau_{shift} \in \{-0.2, -0.4\})$ caused by the shift results in decreased repulsion of the hard true negative samples, delaying convergence.

#### A.3.3 Effect of Learning Rate

As the temperature is decreased, the displacement gradients (Eqn. 5) increase, resulting in an increase in the magnitude of fluctuations from false negative pairs. A low learning rate plays a crucial role in this scenario in smoothening out the fluctuations. On the contrary, at a high learning rate, the fluctuations are amplified and should degrade the performance. However, from Fig. 7, we observe this effect for CIFAR10 only, as the number of false negative pairs in a batch is greater than that in CIFAR100.

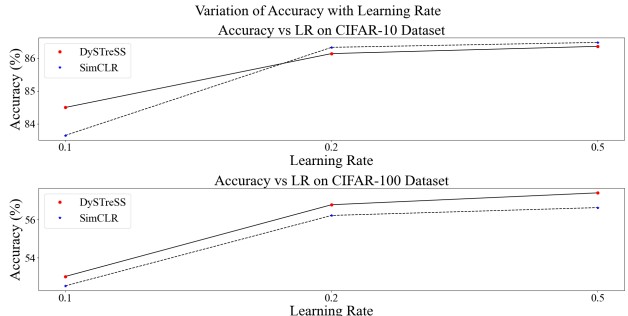

Figure 7: Plot of Accuracy with change in Learning Rate for the datasets CIFAR10 (top) and CIFAR100 (bottom). The figure is best visible at 500%.

## B    EXPERIMENTS ON SENTENCE EMBEDDING

Similar to MACL (Zizheng et al., 2023), we adopt SimCSE (Gao et al., 2021) as the baseline for sentence embedding learning. We conduct experiments with BERT (Devlin et al., 2019) on Semantic Textual Similarity (STS) and Transfer tasks following Gao et al. (2021). We can observe from Table 5 that the proposed framework achieves better performance on most STS and transfer tasks over the SimCSE and MACL baseline under the same experimental environment. In support of the difference in reproduced results from the reported results in the respective papers, we speculate the cause to be the difference in the hardware, as already mentioned in Zizheng et al. (2023). All the experiments were conducted on a 24GB NVIDIA A5500 GPU.

## C    ETIOLOGY OF THE TEMPERATURE FUNCTION

The primary objective of our proposed temperature function is to modulate the temperature for sample pairs to improve representation learning. In SSL, the primary hurdle is the repulsion between the Hard False Negative pairs. Hard True negative pairs also hinder the learning process. In this section, we will discuss the mathematical motivation behind adopting a cosine similarity function as the temperature-modulating function in our work.

**Proposition:**    *The temperature function should have a negative and positive slope in the negative and positive half of the cosine similarity vs. temperature plane, respectively.*

**Proof:**    For negative pairs, the gradient of the InfoNCE loss with respect to $s_{ij}$ will be non-negative, because, if loss decreases, then the cosine similarity of negative pairs should decrease. We assume that the value of this gradient is $\delta$, as shown in Eqn. 9.

$$\frac{\partial \mathcal{L}}{\partial s_{ij}} = \frac{\tau_{ij} - s_{ij}\frac{\partial \tau_{ij}}{\partial s_{ij}}}{\tau_{ij}^2} p_{ij} = \delta \ \text{ where } \delta < \epsilon \ \text{ and } \delta, \epsilon > 0 \tag{9}$$

where $\epsilon$ is a small non-negative number. Expanding the Eqn. 9, we get,

Table 11: Comparison on STS and Transfer tasks (metric: Spearman's correlation with "all" setting). DySTreSS$^+$ means DySTreSS with $\triangle s = -0.2$ and $k = 0.6$, DySTreSS$^*$ means DySTreSS with $\triangle s = -0.4$ and $k = 0.7$, DySTreSS$^{**}$ means DySTreSS with $\triangle s = -0.6$ and $k = 0.8$

| STS Task | STS12 | STS13 | STS14 | STS15 | STS16 | STSB | SICKR | Avg. |
|---|---|---|---|---|---|---|---|---|
| SimCSE (repro.) | 65.10 | 80.32 | 71.42 | 80.51 | 77.80 | 76.47 | 70.69 | 74.62 |
| w/ MACL (repro.) | 67.49 | 81.55 | 73.21 | 80.97 | 77.52 | 76.54 | 70.87 | 74.84 |
| $\tau_{min} = 0.03, \tau_{max} = 0.05$ | | | | | | | | |
| w/ DySTreSS | 68.00 | 81.58 | 73.44 | 80.33 | 77.65 | 76.24 | 70.98 | 74.80 |
| w/ DySTreSS$^+$ | **70.19** | 81.61 | 73.89 | **81.91** | 77.36 | 77.06 | 70.01 | 76.00 |
| w/ DySTreSS$^*$ | 69.34 | 81.76 | 73.51 | 81.61 | 77.39 | 76.67 | 71.53 | 75.96 |
| w/ DySTreSS$^{**}$ | 69.65 | 79.18 | 72.37 | 80.79 | 76.83 | 75.59 | 71.43 | 75.12 |
| $\tau_{min} = 0.02, \tau_{max} = 0.05$ | | | | | | | | |
| w/ DySTreSS$^*$ | 69.17 | **81.69** | 73.73 | 81.38 | 77.75 | 76.19 | 70.98 | 75.84 |
| $\tau_{min} = 0.05, \tau_{max} = 0.07$ | | | | | | | | |
| w/ DySTreSS$^*$ | 69.02 | 80.61 | 73.81 | 81.39 | **79.75** | **77.97** | **72.05** | **76.37** |
| $\tau_{min} = 0.04, \tau_{max} = 0.05$ | | | | | | | | |
| w/ DySTreSS$^*$ | 69.68 | 80.33 | **74.27** | 80.55 | 76.89 | 75.95 | 70.82 | 75.50 |
| Transfer Task | MR | CR | SUBJ | MPQA | SST2 | TREC | MRPC | Avg. |
| SimCSE (repro.) | 80.42 | 85.80 | 94.13 | 88.66 | 85.17 | 87.40 | 72.29 | 84.84 |
| w/ MACL (repro.) | 80.50 | 85.41 | 94.20 | 89.22 | 85.06 | 90.40 | **75.59** | 85.77 |
| $\tau_{min} = 0.03, \tau_{max} = 0.05$ | | | | | | | | |
| w/ DySTreSS | 81.30 | 87.10 | 94.79 | 89.15 | 86.16 | 88.20 | 72.35 | 85.58 |
| w/ DySTreSS$^+$ | **81.72** | 86.76 | 94.34 | 88.31 | 86.33 | 88.00 | 73.74 | 85.60 |
| w/ DySTreSS$^*$ | 81.69 | **87.28** | 94.80 | 89.29 | 85.83 | **91.20** | 72.93 | 86.15 |
| w/ DySTreSS$^{**}$ | 81.62 | 86.68 | 94.74 | 88.75 | **86.71** | 90.40 | 75.30 | **86.31** |
| $\tau_{min} = 0.02, \tau_{max} = 0.05$ | | | | | | | | |
| w/ DySTreSS$^*$ | 81.59 | 86.99 | 94.45 | **89.48** | 86.66 | 87.80 | 74.90 | 85.98 |
| $\tau_{min} = 0.05, \tau_{max} = 0.07$ | | | | | | | | |
| w/ DySTreSS$^*$ | 80.69 | 85.94 | 94.27 | 89.16 | 84.84 | 89.20 | 74.61 | 85.53 |
| $\tau_{min} = 0.04, \tau_{max} = 0.05$ | | | | | | | | |
| w/ DySTreSS$^*$ | 81.02 | 86.23 | **94.97** | 88.86 | 85.17 | 89.80 | 73.22 | 85.61 |

$$\frac{\tau_{ij} - s_{ij}\frac{\partial \tau_{ij}}{\partial s_{ij}}}{\tau_{ij}^2} p_{ij} = \delta$$

$$\implies \frac{\tau_{ij} - s_{ij}\frac{\partial \tau_{ij}}{\partial s_{ij}}}{\tau_{ij}^2} = \frac{\delta}{p_{ij}}$$

$$\implies \tau_{ij} - s_{ij}\frac{\partial \tau_{ij}}{\partial s_{ij}} = \tau_{ij}^2 \frac{\delta}{p_{ij}} \tag{10}$$

$$\implies \frac{\partial \tau_{ij}}{\partial s_{ij}} = \frac{1}{s_{ij}}\left[\tau_{ij} - \tau_{ij}^2 \frac{\delta}{p_{ij}}\right]$$

$$\implies \frac{\partial \tau_{ij}}{\partial s_{ij}} = \frac{\tau_{ij}}{s_{ij}}\left[1 - \frac{\tau_{ij}\delta}{p_{ij}}\right]$$

We can assume that $\tau_{ij} > 0$ without loss of generality.

In self-supervised contrastive learning, the temperature should be high for false negatives to prevent too much repulsion. We have discussed the criteria and the motivation behind our temperature function in Sec. 4.3 of the main manuscript. Also, the temperature should not be very small in the regions with highly negative cosine similarity. We assume that the number of false negatives decreases as we move towards the point $s_{ij} = 0.0$. Hence, for the vanilla case, we will consider two regions, (1) $s_{ij} > 0$ and (2) $s_{ij} \leq 0$.

Expanding the expression for $p_{ij}$ in Eqn. 10, we get,

$$\frac{\partial \tau_{ij}}{\partial s_{ij}} = \frac{\tau_{ij}}{s_{ij}} \left[ 1 - \frac{\tau_{ij}\delta}{\frac{exp(s_{ij}/\tau_{ij})}{\sum_{k=1}^{N} exp(s_{ik}/\tau_{ik})}} \right]$$

$$\implies \frac{\partial \tau_{ij}}{\partial s_{ij}} = \frac{\tau_{ij}}{s_{ij}} \left[ 1 - \frac{\tau_{ij}\delta \sum_{k=1}^{N} exp(s_{ik}/\tau_{ik})}{exp(s_{ij}/\tau_{ij})} \right]$$

$$\implies \frac{\partial \tau_{ij}}{\partial s_{ij}} = \frac{\tau_{ij}}{s_{ij}} \left[ 1 - \tau_{ij}\delta \frac{exp(s_{ij}/\tau_{ij}) + \sum_{\substack{k=1 \\ k \neq j}}^{N} exp(s_{ik}/\tau_{ik})}{exp(s_{ij}/\tau_{ij})} \right]$$

$$\implies \frac{\partial \tau_{ij}}{\partial s_{ij}} = \frac{\tau_{ij}}{s_{ij}} \left[ 1 - \tau_{ij}\delta \frac{exp(s_{ij}/\tau_{ij}) + \sum_{\substack{k=1 \\ k \neq j}}^{N} exp(s_{ik}/\tau_{ik})}{exp(s_{ij}/\tau_{ij})} \right] \qquad (11)$$

$$\implies \frac{\partial \tau_{ij}}{\partial s_{ij}} = \frac{\tau_{ij}}{s_{ij}} \left[ 1 - \tau_{ij}\delta \left( 1 + \frac{\sum_{\substack{k=1 \\ k \neq j}}^{N} exp(s_{ik}/\tau_{ik})}{exp(s_{ij}/\tau_{ij})} \right) \right]$$

$$\implies \frac{\partial \tau_{ij}}{\partial s_{ij}} = \frac{\tau_{ij}}{s_{ij}} \left[ 1 - \tau_{ij}\delta \left( 1 + K \cdot exp(-\frac{s_{ij}}{\tau_{ij}}) \right) \right]$$

where $K = \sum_{\substack{k=1 \\ k \neq j}} exp(\frac{s_{ik}}{\tau_{ik}})$ is taken as a constant with respect to $s_{ij}$, that is, $\frac{\partial K}{\partial s_{ij}} = 0$.

If $N \to \infty$ or for very large N, we can safely assume

$$\frac{exp(s_{ij}/\tau_{ij}) + \sum_{\substack{k=1 \\ k \neq j}}^{N} exp(s_{ik}/\tau_{ik})}{exp(s_{ij}/\tau_{ij})} \simeq \frac{\sum_{\substack{k=1 \\ k \neq j}}^{N} exp(s_{ik}/\tau_{ik})}{exp(s_{ij}/\tau_{ij})} = K \cdot exp(-\frac{s_{ij}}{\tau_{ij}}) \qquad (12)$$

Hence, Eqn. 11 reduces to,

$$\frac{\partial \tau_{ij}}{\partial s_{ij}} = \frac{\tau_{ij}}{s_{ij}} \left[ 1 - \tau_{ij}\delta \left( K \cdot exp(-\frac{s_{ij}}{\tau_{ij}}) \right) \right] \qquad (13)$$

Solving the first-order nonlinear ordinary differential equation given by Eqn. 13, we get,

$$\tau_{ij} = \frac{s_{ij}}{\log(\delta \cdot K \cdot s_{ij} - c)} \qquad (14)$$

where $c$ is the integral constant.

To find the value of $c$, we have to solve for the value of $\tau_{ij}$ at the endpoints of the cosine similarity line. It is to be remembered, $\tau_{ij}$ takes the value $\tau_{max}$ at $s_{ij} = -1$ and $s_{ij} = +1$ (Please refer to Sec. 4.3 in the main manuscript).

Solving, the above equation for the two above-mentioned cases, we get,

$$c^- = -\delta \cdot K - exp(-1/\tau_{max})$$
$$c^+ = -\delta \cdot K - exp(1/\tau_{max}) \qquad (15)$$

Varying the value of the constant in the range $[c^-, c^+]$, we get different curves with different slopes for different values of $\delta$ and $K$, as shown in the Fig. 8

We can observe, that the plotted curves in Fig. 8 do in fact show positive and negative gradients on the positive and negative half of the cosine similarity vs. temperature plane, respectively, as stated in the Sec. 4.3 of the main manuscript. This establishes our theoretically derived condition for the slopes of the temperature function.

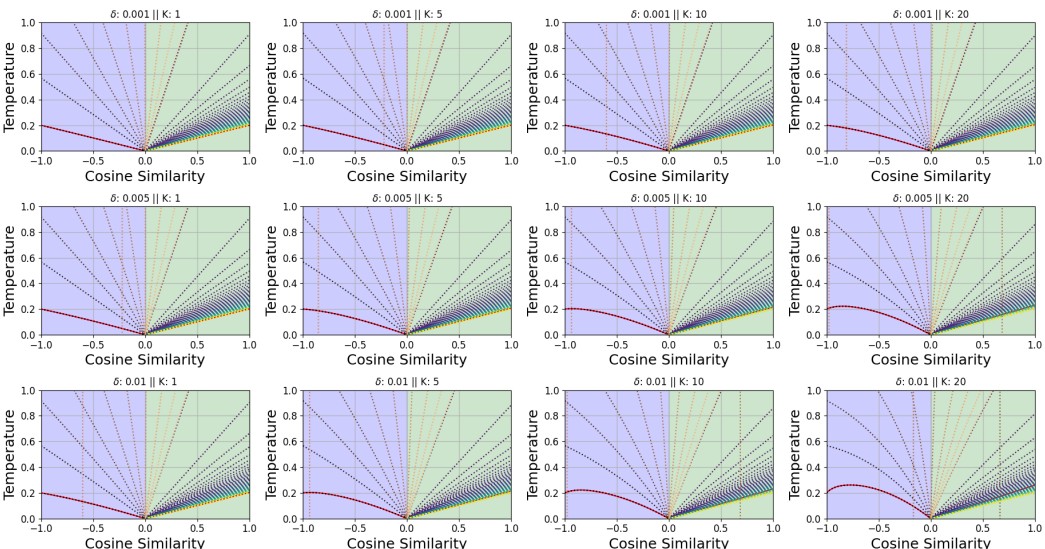

Figure 8: Plots of the solution of ODE in Eqn. 14 for different values of the integral constant, over different values of $\delta$ and $K$.

## D  PSEUDOCODE FOR VANILLA DYSTRESS

---

**Algorithm 3:**  PyTorch-style pseudocode for Vanilla DySTreSS

---

```
# f:  Encoder Network, N: Batch Size, XEnt:  Cross Entropy Loss
# tmin, tmax:  Minimum Temperature, Maximum Temperature
for x in loader:
    # Augment to generate positive pairs
    x_1, x_2 = augment(x)
    # Pass through the encoder to get feature vectors
    z_1, z_2 = f(x_1), f(x_2)
    # L2-Normalize the feature vectors
    z_1, z_2 = F.normalize(z_1, dim=-1), F.normalize(z_2, dim=-1)
    # Generate Cosine Similarity matrix
    s11 = z_1 @ z_1.T
    s22 = z_2 @ z_2.T
    s12 = z_1 @ z_2.T
    s21 = z_2 @ z_1.T
    # Generate Similarity matrix mask
    dg, ndg = torch.eye(N), 1 - torch.eye(N)
    # Segregate the positive pair cosine similarity values
    s11, s22 = s11[ndg].view(N,-1), s22[ndg].view(N,-1)
    # Segregate the negative pair cosine similarity values
    s1211 = torch.cat([s12, s11], dim=1)
    s2122 = torch.cat([s21, s22], dim=1)
    s = torch.cat([s1211, s2122], dim=0)
    temp = tmin + 0.5(tmax - tmin)(1 + cos(π(1 + s.detach())))
    s = s / temp
    labels = torch.arange(N).repeat(2)
    # Compute Cross Entropy loss
    loss = XEnt(logits, labels)
    # Optimize Loss
    loss.backward()
    optimizer.step()
```

---

# E    UNIFORMITY AND TOLERANCE

$$\text{Uniformity} = \log \mathbb{E}_{(x_i, x_j) \sim p_{pair}} \exp\left(-2\|x_i - x_j\|^2\right) \tag{16}$$

$$\text{Tolerance} = -\mathbb{E}_{(x_i, x_j) \sim p_{pos}} < x_i, x_j > = \mathbb{E}_{(x_i, x_j) \sim p_{pos}} \left[-1 + \frac{1}{2}\|x_i - x_j\|^2\right] \tag{17}$$

$$\text{Alignment} = \mathbb{E}_{(x_i, x_j) \sim p_{pos}} \|x_i - x_j\|_2^2 \tag{18}$$

where, $p_{pair}$ and $p_{pos}$ is joint the distribution of all pairs and positive pairs. Hence, uniformity and tolerance (and alignment also) behave the opposite way.

$$\text{Interclass Uniformity} = \log \mathbb{E}_{(x_i, x_j) \sim p_{centroids}} \exp\left(-2\|x_i - x_j\|^2\right) \tag{19}$$

where, $p_{centroids}$ denote the joint distribution of class centroids. A high Interclass Uniformity means that the clusters of the classes are close to each other. Hence, the lower the better.

In Table 12, we provide the overall uniformity and inter-class uniformity of SimCLR, MoCov2, DCL, and DCLW along with our proposed framework. These values are obtained after pretraining on CIFAR10, CIFAR100, and ImageNet1K datasets. Please note that the reported values are for pre-training of 200 epochs, 200 epochs, and 100 epochs for CIFAR10, CIFAR100, and ImageNet1K datasets, respectively. For other epoch numbers, these values are subject to change as the feature representations evolve depending on the convergence of the algorithm.

Table 12: Uniformity and Inter-Class Uniformity metric values for CIFAR10, CIFAR100 and ImageNet1K datasets.

| Method | Uniformity | Inter-Class Uniformity | Accuracy |
|---|---|---|---|
| CIFAR10 | | | |
| SimCLR | -2.2135 | -0.5194 | 83.65 |
| MoCov2 | -2.1929 | -0.5094 | 83.67 |
| DCLW | -2.5482 | -0.5349 | 84.02 |
| DCL | -2.7717 | -0.5524 | 84.47 |
| DySTreSS | -2.8430 | -0.5797 | 85.83 |
| CIFAR100 | | | |
| SimCLR | -2.2556 | -0.5246 | 52.32 |
| MoCov2 | -2.2202 | -0.5364 | 54.01 |
| DCLW | -2.6865 | -0.6405 | 55.87 |
| DCL | -2.8314 | -0.6792 | 56.02 |
| DySTreSS | -2.8169 | -0.7657 | 56.72 |
| ImageNet1K | | | |
| SimCLR | -2.9084 | -0.8739 | 63.2 |
| DCLW | -3.0137 | -0.9544 | 64.2 |
| DCL | -3.0786 | -1.0053 | 65.1 |
| DySTreSS | -3.0596 | -1.0015 | 65.2 |

A lower uniformity value indicates that the samples are more spread out than a model with a higher uniformity value. The inter-class uniformity metric value indicates how far apart the class centroids are from one another. Hence, a lower inter-class uniformity value indicates that the clusters are more separated than a mode with a higher value. We see that the overall uniformity values are the lowest for our proposed method. In fact, we can see a trend in the uniformity values with the accuracy for different models as well. In our case, the inter-class uniformity metric is the lowest, which means that the class centroids are farther away than the other models, and is an indicator of how easily separable the clusters are for our proposed method. Thus, we can say that the proposed framework improves feature representations, and hence, learns more separable clusters, resulting in higher 200-NN accuracy.

# F    GRADIENT OF LOSS WITH RESPECT TO $z_i$

When differentiating the infoNCE loss with respect to a feature vector $z_i$, we need to consider two things, (1) the term where $z_i$ is the anchor, and (2) all the terms where $z_i$ is not the anchor. This gives rise to the two terms in the first line of the differentiation in Eqn. 20.

$$
\begin{aligned}
\frac{\partial \mathcal{L}}{\partial z_i} &= \left[ -\frac{z_{i+}}{\tau} + \frac{\frac{z_{i+}}{\tau} \cdot e^{C_{ii+}} + \sum_{\substack{j=1 \\ j\neq i}}^{N} \frac{z_j}{\tau} \cdot e^{C_{ij}}}{e^{C_{ii+}} + \sum_{\substack{j=1 \\ j\neq i}}^{N} e^{C_{ij}}} \right] + \sum_{\substack{j=1 \\ j\neq i}}^{N} \frac{\frac{z_j}{\tau} \cdot e^{C_{ji}}}{e^{C_{jj+}} + \sum_{\substack{k=1 \\ k\neq j}}^{N} e^{C_{jk}}} \\
&= -\left[ \frac{z_{i+}}{\tau} \left(1 - p^{ii^+}\right) - \sum_{\substack{j=1 \\ j\neq i}}^{N} \frac{z_j}{\tau} \left(p^{i\downarrow j} + p^{j\downarrow i}\right) \right]
\end{aligned}
\tag{20}
$$

# G    ABLATION ON DIFFERENT TEMPERATURE FUNCTIONS

In this section, we present the performance of different temperature functions, namely linear and exponential on CIFAR datasets. We see that all the temperature functions perform similarly to the cosine function, if not better. All the temperature functions satisfy the conditions of positive and negative slope on the positive and negative half of the temperature vs. cosine-similarity plane, respectively. All experiments were run for 200 epochs and the results reported are for 200-NN Top-1 accuracy

Table 13: Comparison of performance on CIFAR datasets for different temperature functions satisfying the conditions in Proposition 1.

| Function | CIFAR10 | CIFAR100 |
|---|---|---|
| Cosine | 85.85 | 56.57 |
| Linear | 85.74 | 56.78 |
| Exponential | 85.81 | 56.47 |

# H    DO WE REALLY NEED HIGH TEMPERATURE IN LOW COSINE SIMILARITY REGION?

In this section, we explore if we need the temperature values to be high in the low cosine similarity region. We experimented with monotonous cosine functions with minimum at $s_{ij} = -1$ and maximum at $s_{ij} = +1$ on the CIFAR datasets. We observed, that using a monotonous cosine function did not degrade performance on the benchmark datasets. From the observed results in Table 14, we can infer that the distribution of false negative samples does not extend towards $s_{ij} = -1$. Also, we can say that, a low temperature on the true negative pairs with cosine similarity in the neighborhood of $s_{ij} = -1$, does not contribute much to the representation learning. This conclusion agrees with the findings of Wang & Liu (2021b). Hence, we do not see any major effect on the performance.

Like easy negative samples, we also consider the possibility of samples that form hard false negative pairs (samples in negative pairs belonging to the same ground truth classes). In the pre-training stage, we can't have access to the ground truth labels, hence cannot say for certain the range of cosine similarities for false negative pairs in the pre-training stage where the weights are initialized randomly. Also, we wouldn't want the samples to drift farther away from each other and hence, we keep a higher temperature towards low cosine similarity.

Table 14: Comaprison of performance on CIFAR datasets for monotonous temperature functions.

| Function | CIFAR10 | CIFAR100 |
|---|---|---|
| Proposed Cosine | 85.85 | 56.57 |
| Monotonic Cosine | 85.84 | 56.21 |

# I COMPARISON WITH OTHER FRAMEWORKS

In this section, we present the comparison of the performance of our proposed framework with DINO and WMSE on CIFAR datasets. All experiments were done with a batch size of 256 and trained for 200 epochs. The comparison results on CIFAR10 and CIFAR100 datasets are presented in Table 15.

Table 15: Comparison with DINO and WMSE on CIFAR datasets

| Methods | CIFAR10 | CIFAR100 |
|---|---|---|
| DINO | 84.02 | 46.79 |
| WMSE | 85.52 | 52.72 |
| DySTreSS | **85.68** | **56.57** |

For the ImageNet100 dataset, we ran pre-training for 400 epochs with a batch size of 128. We used the same training conditions as in Zhang et al. (2022b) and Zhang et al. (2022a), for a fair comparison. From the results presented in Table 16. obtained on the ImageNet100 dataset, we can see that our proposed method clearly outperforms the state-of-the-art methods like DINO (Caron et al., 2021), WMSE (Ermolov et al., 2021), Zero-CL (Zhang et al., 2022b), and ARB (Zhang et al., 2022a).

Table 16: Comparison of the proposed method with DINO, WMSE, Zero-CL, and ARB on the ImageNet100 dataset (Accuracy for WMSE is taken from Zhang et al. (2022b) and Zhang et al. (2022a))

| Methods | Projector Dimension | Top-1 Linear Eval. Accuracy (%) |
|---|---|---|
| Barlow Twins (Zbontar et al., 2021) | 2048 | 78.62 |
| SimSiam (Chen & He, 2021) | 2048 | 77.04 |
| DINO (Caron et al., 2021) | 2048 | 74.84 |
| WMSE (Ermolov et al., 2021) | 256 | 69.06 |
| ZeroCL (Zhang et al., 2022b) | 2048 | 79.26 |
| ZeroFCL (Zhang et al., 2022b) | 2048 | 79.32 |
| ARB (Zhang et al., 2022a) | 2048 | 79.48 |
| DySTreSS (OURS) | 2048 | **81.24** |

