# OpenReview forum: "DySTreSS: Dynamically Scaled Temperature in Self-Supervised Contrastive Learning"
_ICLR.cc/2024/Conference — Submitted to ICLR 2024_

### Official Review · Reviewer_RTHa · 2023-10-20

**Soundness:** 2 fair
**Presentation:** 2 fair
**Contribution:** 2 fair
**Rating:** 6
**Confidence:** 3

**Summary:**

This paper intends to control the temperature to improve the performance of the self-supervised contrastive learning in image classification, long-tailed image classification and semantic textual similarity estimation. Specifically, author propose to compute sample-wise temperature in the consideration of cosine-similarity between data pairs. The proposed methodology prevents hard negative samples from being repulsed away, thereby improving the performance of the baseline method (SimCLR).

**Strengths:**

1) This paper proposes a novel dynamic temperature scaling method for self-supervised contrastive learning and reports improved performance.
2) The paper shows that the proposed method can improve performance on a variety of tasks, including image classification in long-tailed datasets, small-scale benchmarks, and semantic textual similarity (SES).
3) The paper honestly cites related works related to core concepts ([1], [2]) and sufficiently shows that the uniformity-tolerance trade-off caused by temperature scaling is an important concept that has been discussed in many papers.
4) The paper well proves the necessity of dynamic temperature scaling through mathematical intuition.


[1] Feng Wang and Huaping Liu. Understanding the behavior of contrastive loss

[2] Tongzhou Wang and Phillip Isola. Understanding contrastive representation learning through alignment and uniformity on the hypersphere. In ICML, 2020.

**Weaknesses:**

1) The paper lacks explanations of key concepts. For example, to understand this paper, it is important to understand what FN, PN, and TP mean, but the paper does not kindly define each of them.
2) Some experimental results are omitted. For example, the paper states in the second paragraph of page 4 that it was inspired by the EMD ratio of TN and FN, but the experimental results are not reported.
3) There are some coarse parts of the notation. For example, the sample index is written as a superscript in equation 5, but not in other equations. In addition, p^(i↓j)+p^(ij↓) seems to be a typo of p^(i↓j)+p^(j↓i). In addition, ∂L/(∂s_ij )>0=δ in equation 7 should be changed to ∂L/(∂s_ij )=δ>0.
4) The theoretical analysis seems a bit strange. First, the proposition contains information about the slope of the temperature function, but this is not a mathematically proven fact, but just about the idea of the proposed algorithm. In addition, the author's claim that ∂L/(∂s_ij ) in equation 8 is always negative is suspicious.
5) Originally, only one temperature hyperparameter needs to be tuned for existing contrastive learning, but the proposed DySTress has more hyperparameters (τ_min,τ_max,Δ_s,k) as a result, and it can be seen from Table 6 that the performance of DySTress is sensitive to the values of hyperparameters.

**Questions:**

1) True negatives with high cosine similarity may actually belong to different classes. However, could DySTreSS hinder the proper repulsoin from those TNs, thereby accumulating noises and aggravating overfitting?
2) Why is the temperature scaling function used in the main text different from the temperature scaling function obtained from the ODE solution in the appendix? And why was the function in Algorithm 1 chosen as the temperature scaling function among several candidates? Are there any comparison experiments with other temperature scaling functions?
3) In the Experimental Results section, the proposed method seems to have been carefully tuned for optimal performance w,r,t tge hyperparameters (Δ_s,k, etc.). Were the hyperparameters for the other baseline methods also tuned in a fair manner?
4) Cosine similarity is high for hard negative samples, so it makes sense to give them a high temperature to reduce their weight. However, why do we need to increase the temperature for easy negative samples with low cosine similarity? I think a monotonically increasing temperature function would make more sense. Are there any experiments with this?

---

> ### Author Response · Authors · 2023-11-19
> **Reply to Reviewer RTHa (1/3)**
>
> Thanks to the reviewer for acknowledging our strenghts:
> - proposes a novel dynamic temperature scaling method for self-supervised contrastive learning and reports improved performance.
> - shows that the proposed method can improve performance on a variety of tasks, including image classification in long-tailed datasets, small-scale benchmarks, and semantic textual similarity (SES).
> - honestly cites related works related to core concepts ([1], [2]) and sufficiently shows that the uniformity-tolerance trade-off caused by temperature scaling is an important concept that has been discussed in many papers.
> - proves the necessity of dynamic temperature scaling through mathematical intuition.
>
> ---
>
> > [W1] The paper lacks explanations of key concepts. For example, to understand this paper, it is important to understand what FN, PN, and TP mean, but the paper does not kindly define each of them.
>
> We thank the reviewer for this question. We would like to bring the attention of the reviewer to Sec. 3 (below Eqn. 1), where we have already defined True Positive (TP), False Negative (FN) and True Negative (TN) pairs. In the context of contrastive learning, every sample belongs to a class of its own, since we do not have any ground truth label information. In the pre-training stage, we can only say if a pair is positive or negative. However, when dealing with all these pairs, we can argue that there are samples which may belong to the same ground truth class but could be in a negative pair. These pairs are False Negative (FN) pairs. Similarly, there are pairs which contain samples obtained by different augmentations of the same sample, and we term these pairs as True Positive (TP) Pairs. Lastly, pairs containing samples which may belong to different ground truth classes are termed as True Negative (TN) pairs.
>
> ---
>
> > [W2] Some experimental results are omitted. For example, the paper states in the second paragraph of page 4 that it was inspired by the EMD ratio of TN and FN, but the experimental results are not reported.
>
> We would like to draw the attention of the reviewer to the fact that we merely studied the EMD between the TN and FN pairs to see how the pairs behave before and after pre-training. However, we do mention the magnitude of the EMD metric between TN and FN pairs in text in the same paragraph.
>
> ---
>
> > [W3] There are some coarse parts of the notation. For example, the sample index is written as a superscript in equation 5, but not in other equations. In addition, p^(i&darr;j)+p^(ij&darr;) seems to be a typo of p^(i&darr;j)+p^(j&darr;i). In addition, $\partial$L/($\partial$s_ij )>0= $\delta$ in equation 7 should be changed to $\partial$L/($\partial$s_ij )=$\delta$>0.
>
> Thanks for the suggestion. We have made the necessary changes in the manuscript.
>
> ---
>
> > [W4] The theoretical analysis seems a bit strange. First, the proposition contains information about the slope of the temperature function, but this is not a mathematically proven fact, but just about the idea of the proposed algorithm. In addition, the author's claim that $\partial L/(\partial s_{ij} )$ in equation 8 is always negative is suspicious.
>
> We prove the proposition in Appendix C, where we derive the nature of the slope of the temperature function in detail. On plotting the derived temperature function in Equation 14, we plot the curve in Fig. 8, and we can observe the nature of the temperature function from that figure itself.
>
> Furthermore, we would like to draw the attention of the reviewer to the fact that for the negative pairs $\partial L/(\partial s_{ij} )$ is a “non-negative” quantity.  We had also mentioned it in the manuscript. In fact, the reviewer has also kindly suggested a correction to the order of ‘>0’ and ‘=$\delta$’ in the previous comment for the same.
>
> ---

---

> ### Author Response · Authors · 2023-11-19
> **Reply to Reviewer RTHa (2/3)**
>
> > [W5] Originally, only one temperature hyperparameter needs to be tuned for existing contrastive learning, but the proposed DySTress has more hyperparameters ($\tau_{min}$,$\tau_{max}$,$\Delta s$,k) as a result, and it can be seen from Table 6 that the performance of DySTress is sensitive to the values of hyperparameters.
>
> Thanks to the reviewer for pointing this out. From Table 6, it can be seen how the performance varies with the parameters $\tau_{min}$ and $\tau_{max}$. However, we have already provided an intuitive explanation why this happened in Sec. 4.2, that is, the effect of temperature on the local and global structures in the feature space. Table 6 only proves the statements made in Sec. 4.2. The general approach we took when applying DySTreSS on top of SimCLR, is that we set $\tau_{max}$ to the optimal temperature for SimCLR and tuned only $\tau_{min}$ to understand how far we can decrease $\tau_{min}$ before performance starts to degrade.
>
> The hyper-parameter $\Delta s$ is to study and get an idea about the optimal temperature function, as we have previously stated in Sec. 4.3, that the boundary between false negative pairs and true negative pairs is not hard. However, we see in Table 7, that the performance is not sensitive to the value of $\Delta s$, and may be ignored or not tuned. The hyper-parameter k is to maintain the minimum and maximum temperature value $\tau_{min}$ and $\tau_{max}$, respectively, hence, can be ignored as a hyper-parameter, as it does not require tuning and is a byproduct of using $\Delta s$.
>
> ---
>
> > [Q1] True negatives with high cosine similarity may actually belong to different classes. However, could DySTreSS hinder the proper repulsoin from those TNs, thereby accumulating noises and aggravating overfitting?
>
> We have actually used the terminology True Negatives pairs for the pairs which consist of samples from different classes. If those true negative pairs have high cosine similarities then applying a high temperature can attenuate the repulsion. However, we have to keep in mind that false negative pairs (negative pairs, but with samples from the same class) can also lie in the high cosine similarity region. This can lead to increased repulsion between samples in the false negative pairs, thereby mapping samples from the same class farther away and degrading performance as can be seen from the results presented in Table 6 of the manuscript.
>
> We hope this explanation is clear. Otherwise, we would like to request the reviewer to elaborate the question.
>
> ---
>
> > [Q2] Why is the temperature scaling function used in the main text different from the temperature scaling function obtained from the ODE solution in the appendix? And why was the function in Algorithm 1 chosen as the temperature scaling function among several candidates? Are there any comparison experiments with other temperature scaling functions?
>
> Due to the presence of logarithm in the denominator, logarithm can assume value tending to zero. This will lead to the value of $\tau_{ij}$ tending to infinity.Due to the possibility of numerical instability, we opted for different forms of temperature function but with the properties that we have mentioned in the paper, that is, negative and positive slope in the negative and positive half of the cosine similarity vs temperature plane. We experimented with different types of functions, like linear and exponential. However we observed that we obtained similar performance with the cosine function. The accuracies are given in the table below, and will be added in the appendix of the manuscript. The corresponding codes will also be added in the github repository.
>
> Table 1.
> | Function Type | CIFAR10 | CIFAR100 |
> |---|---|---|
> |Cosine|85.85|56.57|
> |Linear|85.74|56.78|
> |Exponential|85.81|56.47|
>
> However, using the exponential function involves setting up two parameters to maintain the maximum and minimum temperature at $\tau_{max}$ and $\tau_{min}$, respectively, which needs to be manually tuned and verified. Whereas, it is easier for the cosine function to maintain the same. Although the linear function has no such extra parameters, both the linear and exponential function will be non-differentiable at $s_{ij} = 0.0$ (Equation 11). Hence, we argue that use of the cosine function is a better choice.
>
> ---
>
> > [Q3] In the Experimental Results section, the proposed method seems to have been carefully tuned for optimal performance w,r,t tge hyperparameters ($\Delta s$,k, etc.). Were the hyperparameters for the other baseline methods also tuned in a fair manner?
>
> Yes, the results for the other baseline methods used in the manuscript were the best that can be obtained using the lightly-ai library that we used for all our experiments. The same values are reported in the official website of lightly-ai. For the ImageNet experiments, we used the original hyper-parameters from the paper and implemented then using the lightly-ai library.
>
> ---

---

> ### Author Response · Authors · 2023-11-19
> **Reply to Reviewer RTHa (3/3)**
>
> > [Q4] Cosine similarity is high for hard negative samples, so it makes sense to give them a high temperature to reduce their weight. However, why do we need to increase the temperature for easy negative samples with low cosine similarity? I think a monotonically increasing temperature function would make more sense. Are there any experiments with this?
>
> Primarily, our motivation of using the cosine function as the temperature function comes from the derived form in Appendix C. However, we respect the suggestion from the reviewer and ran our proposed framework with a monotonically increasing temperature function.
>
> Like easy negative samples, we also consider the possibility of samples that form hard false negative pairs (samples in negative pairs belonging to the same ground truth classes). In the pre-training stage, we can’t have access to the ground truth labels, hence cannot say for certain the range of cosine similarities for false negative pairs in the pre-training stage where the weights are initialized randomly. Also, we wouldn't want the samples to drift farther away from each other and hence, we keep a higher temperature towards low cosine similarity.
>
> We have included the results in Table 2 below, for monotonically increasing functions in the table below (as well as in the Appendix of the manuscript). We see that the results are almost similar to the reported results, which does in fact give a crude idea about the range of cosine similarity of the hard false negative pairs resulting from augmentations applied to positive pairs. It also proves that the easy true negative pairs do not contribute or have little effect on representation learning. This conclusion agrees with the hypothesis stated in [1].
>
> We have included the above discussion in Appendix H in the revised manuscript.
>
> Table 2. Comparison of performance on CIFAR datasets when using proposed cosine function and monotonic function as the temperature function in the DySTreSS framework.
>
> | Type of Cosine | CIFAR10 | CIFAR100 |
> |---|---|---|
> | Proposed Cosine | 85.85 | 56.57 |
> | Monotonic Cosine | 84.57 | 55.61 |
>
> ---
>
> References:
>
> [1] Feng Wang and Huaping Liu. Understanding the behavior of contrastive loss, CVPR 2021

---

> > ### Comment · Reviewer_RTHa · 2023-11-22
> > **Post Rebuttal**
> >
> > Dear authors,
> >
> > It is much better to see that the authors clarify the original contributions of this paper.
> >
> > After reading the response and Appendix.C , I understand the nature of the slope of the temperature function in detail.
> >
> > I am glad to see the authors have investigated various function types including monotonic Cosine, Linear and Exponential.
> >
> > Overall, I think the authors have extensively addressed my concerns.
> >
> > Therefore, I am happy to increase my original score.

---

> > > ### Author Response · Authors · 2023-11-23
> > > **Reply to Reviewer RTHa**
> > >
> > > Sincere thanks for your positive comments.

---

### Official Review · Reviewer_iD3A · 2023-10-23

**Soundness:** 2 fair
**Presentation:** 3 good
**Contribution:** 1 poor
**Rating:** 5
**Confidence:** 5

**Summary:**

This paper proposes DySTreSS to improve the performance of InfoNCE loss in self-supervised contrastive learning methods, especially for negative-required CL methods. Then, the author provides a mathematical analysis to support the construction of the proposed dynamically scaled temperature function. However, Experimental results show limited improvements over the baselines. Besides, due to the limitation of negative-required methods (batch size sensitive), and the mainstream of CL is the negative-free framework (DINO, iBOT), the significance of exploring NCE loss is somewhat limited.

**Strengths:**

The paper is well-written and easy to follow.

The proposed method is sound and the experimental results show a few improvements over the baselines.

**Weaknesses:**

1. As stated in the summary, due to the limitation of negative-required methods (batch size sensitive), and the mainstream of CL is the negative-free framework (DINO, iBOT), the significance of exploring NCE loss is somewhat limited.

2. The improvements are limited, which can be obtained by switching hyperparameters (e.g., lr, wd). Could the author provide the mean and variance over 5 times running?

3. The accuracies reported in Table 2 are too low. For example, Barlow Twins with 100 epoch pretraining can actually achieve 67+ top-1 accuracy. However, only 62.9 is reported in Table 2. Perhaps the author should tune the hyperparameters to make the proposed method comparable.

4. missing references. The authors miss some negative-free methods (e.g., Zero-CL, W-MSE, DINO, ARB) and should compare with them the main results (as they also provide results on ResNets). I recommend the author discuss the advantages and disadvantages of the proposed methods and these negative-free methods.

[1] Zhang S, Zhu F, Yan J, et al. Zero-cl: Instance and feature decorrelation for negative-free symmetric contrastive learning[C]//International Conference on Learning Representations. 2021.

[2] Caron M, Touvron H, Misra I, et al. Emerging properties in self-supervised vision transformers[C]//Proceedings of the IEEE/CVF international conference on computer vision. 2021: 9650-9660.

[3] Ermolov A, Siarohin A, Sangineto E, et al. Whitening for self-supervised representation learning[C]//International Conference on Machine Learning. PMLR, 2021: 3015-3024.

[4] Zhang S, Qiu L, Zhu F, et al. Align representations with base: A new approach to self-supervised learning[C]//Proceedings of the IEEE/CVF Conference on Computer Vision and Pattern Recognition. 2022: 16600-16609.

**Questions:**

See weaknesses.

---

> ### Author Response · Authors · 2023-11-19
> **Reply to Reviewer iD3A (1/3)**
>
> Thanks for acknowledging our strengths:
> - paper is well-written and easy to follow.
> - proposed method is sound and the experimental results show a few improvements over the baselines.
>
> ---
>
> > [W1] As stated in the summary, due to the limitation of negative-required methods (batch size sensitive), and the mainstream of CL is the negative-free framework (DINO, iBOT), the significance of exploring NCE loss is somewhat limited.
>
> Thanks to the reviewer for this comment. The frameworks like DINO [2] and iBOT [7] fall under the canopy of instance-based non-contrastive self-supervised learning frameworks, and utilizes self-distillation with no labels. Although the mainstream focus of CL has shifted to non-contrastive methods, however, the role of temperature in contrastive learning is an intriguing phenomenon and still not explored fully. We can see works like iSogCLR [5], Kukleva et al. [6], DCL [8], DirectCLR [9], etc. where the authors have explored the self-supervised contrastive learning setting in recent years (2022-23). In this work, we also intend to study the effect of temperature in contrastive learning (InfoNCE loss based frameworks) and improve the same. We agree that the contrastive learning frameworks come with its limitations. However, we believe that there is more to the function of temperature hyper-parameter in the InfoNCE loss, than what meets the eye. Carefully tuning the temperature hyper-parameter can greatly change the representation learning capability of the network as evident from the results.
>
> ---
> > [W2] The improvements are limited, which can be obtained by switching hyperparameters (e.g., lr, wd). Could the author provide the mean and variance over 5 times running?
>
> For the CIFAR10 dataset, we ran the proposed method with 5 different random seeds. We obtained a mean accuracy of 85.854 with a standard deviation of 0.0621. For the CIFAR100 dataset, we obtained a mean accuracy of 56.568 with a standard deviation of 0.215.
>
> ---
> > [W3] The accuracies reported in Table 2 are too low. For example, Barlow Twins with 100 epoch pretraining can actually achieve 67+ top-1 accuracy. However, only 62.9 is reported in Table 2. Perhaps the author should tune the hyperparameters to make the proposed method comparable.
>
> We used the lightly-ai library for all our experiments. We ran all our experiments on 24GB NVIDIA A5000 GPU for ImageNet1K, with batch size 256 and FP16 precision. The hyper-parameters for Barlow Twins used were the same as used in the paper, consequently resulting in the reported result. The result can also be seen on the lightly-ai benchmark website. We in fact tried to investigate the performance of Barlow Twins, but obtained the same result as reported there. On the contrary, the reported accuracy for Barlow Twins in the manuscript of Zero-CL is 67.7 for a batch size of 1024.
>
> ---

---

> ### Author Response · Authors · 2023-11-19
> **Reply to Reviewer iD3A (2/3)**
>
> >[W4] missing references. The authors miss some negative-free methods (e.g., Zero-CL, W-MSE, DINO, ARB) and should compare with them the main results (as they also provide results on ResNets). I recommend the author discuss the advantages and disadvantages of the proposed methods and these negative-free methods.
>
> Thanks for pointing this out and we have considered these now. We have now compared and added a discussion about the advantages and disadvantages of the above mentioned methods in Sec. 2 (Related Works).
>
> In the table below, we have added the performance of DINO and WMSE on CIFAR datasets only with a batch size of 128 and pre-training of 200 epochs.
>
> Table 1. Comparison of the the proposed method with DINO and WMSE on CIFAR datasets
> | Methods | CIFAR10 | CIFAR100 |
> |---|---|---|
> | DINO | 84.02 | 46.79 |
> | WMSE | 85.52 | 52.72 |
> | DySTreSS | 85.68 | 56.57 |
>
> We also ran an experiment on the ImageNet100 dataset for 400 epochs with a batch size of 128, as done in [4]. We observed that it outperformed DINO [2], WMSE [3], ZeroCL [1], and ARB [4] by a satisfactory margin. The evidence is presented in Table 2 below.
>
> Table 2. Comparison of the proposed method with DINO, WMSE, Zero-CL, and ARB on the ImageNet100 dataset (Accuracy for WMSE is taken from [1] and [4])
> | Methods | Projector Hidden Layer Dimension | Top-1 Linear Eval. Accuracy (%) |
> |---|---|---|
> | DINO [2] | 2048 |74.84 |
> | WMSE [3] | 256 | 69.06 |
> | ZeroCL [1] | 2048 | 79.26 |
> | ZeroFCL [1] | 2048 | 79.32 |
> | ARB [4] | 2048 | 79.48 |
> | DySTreSS| 2048 | 81.24 |
>
> The proposed work attempts to study the effect of temperature hyper-parameter in contrastive learning methods and improves the same. The advantage of our proposed method is that it improves the performance of contrastive learning frameworks by tuning the temperature hyper-parameter as a function of cosine similarity. It also outperforms several contrastive learning methods, with the same compute resource and time requirement. Tuning the temperature hyper-parameter for each pair allows the framework to give more or less importance based on the cosine similarity of the samples in the pair. While the infoNCE loss itself takes care of the global semantics, tuning the temperature as a function of cosine similarity, allows the framework to take the local semantics into consideration as well.
>
> However, methods like Zero-CL[1], W-MSE[3] are said to be negative-free contrastive learning frameworks. Although ZeroCL is a negative free method, it does use information from non-positive pairs to compute batch and dimension-wise statistics. W-MSE uses a whitening strategy. W-MSE also uses information from samples in non-positive pairs to compute batch statistics and computing the Whitening matrix with a complexity same as computing a cosine similarity matrix. Furthermore, Cholesky decomposition is done on the Whitening matrix with a complexity of $O(k^3 + Mk^2)$. However, our proposed method does not incur any such extra complexity.
>
> Methods like DINO [2], iBOT [5] use knowledge distillation and require two separate networks and an extra parameter update step for the teacher network. However, our method does not use any separate network and also doesn't use an extra parameter update step. However, knowledge distillation based methods, like DINO and iBOT have shown to outperform contemporary frameworks on benchmark datasets.
>
> ARB [4] proposes a Nearest Orthonormal Basis based optimization objective. The authors claim that the objective complexity is O(d), but to take care of the Non-full rank matrix case, the authors use a pseudo-orthonormal base, which involves computing a spectral decomposition of the correlation matrix. This particular computational complexity is absent in our framework.
>
> ---

---

> > ### Author Response · Authors · 2023-11-21
> > **Reply to Reviewer iD3A (3/3)**
> >
> > References:
> >
> > [1] Zhang S, Zhu F, Yan J, et al. Zero-cl: Instance and feature decorrelation for negative-free symmetric contrastive learning[C]//International Conference on Learning Representations. 2021.
> >
> > [2] Caron M, Touvron H, Misra I, et al. Emerging properties in self-supervised vision transformers. Proceedings of the IEEE/CVF international conference on computer vision. 2021: 9650-9660.
> >
> > [3] Ermolov A, Siarohin A, Sangineto E, et al. Whitening for self-supervised representation learning. International Conference on Machine Learning. PMLR, 2021: 3015-3024.
> >
> > [4] Zhang S, Qiu L, Zhu F, et al. Align representations with base: A new approach to self-supervised learning. Proceedings of the IEEE/CVF Conference on Computer Vision and Pattern Recognition. 2022: 16600-16609.
> >
> > [5] Kukleva et al,. Temperature schedules for self-supervised contrastive methods on long-tail data. In ICLR 2023.
> >
> > [6] Qiu et al,. Not All Semantics are Created Equal: Contrastive Self-supervised Learning with Automatic Temperature Individualization. In ICML 2023.
> >
> > [7] Jinghao Zhou, Chen Wei, Huiyu Wang, Wei Shen, Cihang Xie, Alan Yuille, and Tao Kong. ibot: Image bert pre-training with online tokenizer. International Conference on Learning Representations (ICLR), 2022.
> >
> > [8] Yeh, CH., Hong, CY., Hsu, YC., Liu, TL., Chen, Y., LeCun, Y. (2022). Decoupled Contrastive Learning. In: Avidan, S., Brostow, G., Cissé, M., Farinella, G.M., Hassner, T. (eds) Computer Vision – ECCV 2022.
> >
> > [9] Jing, L., Vincent, P., LeCun, Y., & Tian, Y. (2021). Understanding Dimensional Collapse in Contrastive Self-supervised Learning. ICLR 2022

---

> > > ### Comment · Reviewer_iD3A · 2023-11-22
> > > **The authors' response does not address my concerns.**
> > >
> > > Thanks to the authors for the response. However, I still think the results are not comparable. My main concerns draw into two aspects:
> > >
> > > 1) The low accuracy reported in Table. 2. Although the authors say they use lightly-ai, leading to the results in Table 2. However, in fact, I also ran the official code of Barlow Twins with batch size 256, and I got 67+ top-1 accuracies. Besides, Other conventional methods (e.g., SimSiam) also report 68+ top-1 accuracy under 100 epochs with 256 batch size. Therefore, I believe Table. 2 gives too low accuracies of other compared methods.
> > >
> > > 2) The author only reports KNN and linear probing results, which are not convincing enough for SSL methods. I suggest the author provide fine-tuned results on classification, detection, and segmentation tasks.
> > >
> > > Overall, I tend to reject this paper.

---

> > > > ### Author Response · Authors · 2023-11-22
> > > > **Reply to Reviewer iD3A**
> > > >
> > > > > [1] The low accuracy reported in Table. 2. Although the authors say they use lightly-ai, leading to the results in Table 2. However, in fact, I also ran the official code of Barlow Twins with batch size 256, and I got 67+ top-1 accuracies. Besides, Other conventional methods (e.g., SimSiam) also report 68+ top-1 accuracy under 100 epochs with 256 batch size. Therefore, I believe Table. 2 gives too low accuracies of other compared methods.
> > > >
> > > > We present the [TensorBoard](https://tensorboard.dev/experiment/NxyNRiQsQjWZ82I9b0PvKg/#scalars) logs to the reviewer as a proof. The configuration and comparison results are also available on the lightly-ai benchmark [website](https://docs.lightly.ai/self-supervised-learning/getting_started/benchmarks.html). We do not accept that Barlow Twins with a batch size of 256 gave 67+ top-1 accuracies. We ask the reviewer to share some authentic sources as per the reviewer's guidelines.
> > > >
> > > > We would like to draw the attention of the reviewer to the fact, that the result reported in ZeroCL [1] (Sec. 4, Pg. No. 6), which has an accuracy of Top-1 linear probing accuracy of 67.7%, for a batch size of **1024**.
> > > >
> > > > In the results presented in Table 2 in the reply to the reviewer (and also in Appendix I), the reported accuracy of the proposed method outperforms the accuracy reported by Barlow Twins in ZeroCL [1] or ARB [3], on the ImageNet100 dataset, after pre-training for 400 epochs.
> > > >
> > > > We agree that the accuracy of SimSiam on the ImageNet1K dataset, with a batch size of 256 and 100 epochs pre-training is 68.1 as reported in the SimSiam paper. However, the training conditions differ from our proposed method (different optimizer and learning rate schedule).
> > > >
> > > > However, we would like to point out that, as per the findings in ZeroCL [1] and ARB [3], the reported Top-1 linear probing accuracy of SimSiam on the ImageNet100 dataset after 400 epochs of pre-training, is 77.04%, which is lower than the accuracy obtained by our proposed method (Appendix I). The proposed method (Top-1 200NN accuracy = 85.68%) also outperforms SimSiam (Top-1 200NN accuracy = 81.9%) on the CIFAR10 dataset, after 200 epochs of pre-training and with a batch size of 128.
> > > >
> > > > ---
> > > > > [2] The author only reports KNN and linear probing results, which are not convincing enough for SSL methods. I suggest the author provide fine-tuned results on classification, detection, and segmentation tasks.
> > > >
> > > > We would like to point out that most of the recent works (even the works cited by the reviewer) do not report detection, segmentation, or fine-tuned classification results. We are mentioning only a few of the recent papers below,
> > > > - ZeroCL [1]
> > > > - WMSE [2]
> > > > - ARB [3]
> > > >
> > > > We also want to draw the attention of the reviewer to Table 5 (Sec. 6.4 of the main manuscript) and Table 11 (Appendix B), where we have reported detailed results on Sentence Embedding Similarity tasks. We observed that the proposed algorithm outperforms the baseline SimCSE [4] and MACL [5] on almost all the tasks. Thus, we request the reviewer to reconsider our case, considering these additional tasks.
> > > >
> > > > ---
> > > > References:
> > > >
> > > > [1] Zhang S, Zhu F, Yan J, et al. Zero-cl: Instance and feature decorrelation for negative-free symmetric contrastive learning. International Conference on Learning Representations. 2021.
> > > >
> > > > [2] Ermolov A, Siarohin A, Sangineto E, et al. Whitening for self-supervised representation learning[C]//International Conference on Machine Learning. PMLR, 2021: 3015-3024.
> > > >
> > > > [3] Zhang S, Qiu L, Zhu F, et al. Align representations with base: A new approach to self-supervised learning[C]//Proceedings of the IEEE/CVF Conference on Computer Vision and Pattern Recognition. 2022: 16600-16609.
> > > >
> > > > [4] Tianyu Gao, Xingcheng Yao, and Danqi Chen. SimCSE: Simple contrastive learning of sentence embeddings. In Proceedings of the 2021 Conference on Empirical Methods in Natural Language Processing, pp. 6894–6910
> > > >
> > > > [5] Huang Zizheng, Chen Haoxing, Wen Ziqi, Zhang Chao, Li Huaxiong, Wang Bo, and Chen Chunlin. Model-aware contrastive learning: Towards escaping the dilemmas. In ICML, 2023

---

> ### Comment · Reviewer_iD3A · 2023-11-23
> **Questions about the reproductions.**
>
> Thanks for the authors' efforts and detailed response.
>
> I believe reproducing a prior work, the most convincing way is using their official code [https://github.com/facebookresearch/barlowtwins]. Do the authors run this code and could the authors share the pretraining and evaluation logs? Then, back to SimSiam, the authors argue the training recipe is different, so why not use these popular SSL settings (e.g., LARs, lr scheduler)?
>
> Then, the authors say that "most recent works" do not report detection and segmentation results. However, in my opinion, these methods are not "most recent'", since W-MSE is published on ICML 21, and ARB is published on CVPR 22. Most of the methods [1, 2, 3, 4, 5, 6, 7] in 2023 (especially after the appearance of MIM-based methods) have reported fine-tuning classification, detection, and segmentation results. Therefore, I think requiring the transfer learning results on detection and segmentation is reasonable.
>
> Then, back to the meaning of the initial idea, I still think extensive research on temperature in negative-requiring contrastive methods will not bring too much insight into this community.
>
> [1] Masked Autoencoders Are Scalable Vision Learners, CVPR 2022
> [2] SimMIM: A Simple Framework for Masked Image Modeling, CVPR 2022
> [3] Context Autoencoder for Self-Supervised Representation Learning, IJCV 2023
> [4] Siamese Image Modeling for Self-Supervised Vision Representation Learning, CVPR 2023
> [5] iBOT: Image BERT Pre-Training with Online Tokenizer, ICLR 2022
> [6] MCMAE: Masked Convolution Meets Masked Autoencoders, NeurIPS 2022,
> [7] Patch-level Representation Learning for Self-supervised Vision Transformers, CVPR 2022

---

> > ### Author Response · Authors · 2023-11-23
> > **Reply to Reviewer iD3A**
> >
> > > I believe reproducing a prior work, the most convincing way is using their official code [https://github.com/facebookresearch/barlowtwins]. Do the authors run this code and could the authors share the pretraining and evaluation logs? Then, back to SimSiam, the authors argue the training recipe is different, so why not use these popular SSL settings (e.g., LARs, lr scheduler)?
> >
> > Thank you for your comments. As we mentioned in our manuscript, we used the same library (lightly-ai) to compare all the methods, and our training recipe is same as the official code. This can be verified [here](https://github.com/lightly-ai/lightly/blob/master/benchmarks/imagenet/resnet50/barlowtwins.py).
> >
> > Sorry for the confusion. We **do use** LARS for ImageNet100 and ImageNet1K datasets, and SGD for CIFAR datasets, following recent works [8]. We definitely could run SimSiam given a sufficient time (100 epochs ImageNet1K takes around 7 days on our resources). We wanted to mention that the reported results in SimSiam paper should not be directly compared with our results as the two settings are different.
> >
> > > Then, the authors say that "most recent works" do not report detection and segmentation results. However, in my opinion, these methods are not "most recent'", since W-MSE is published on ICML 21, and ARB is published on CVPR 22. Most of the methods [1, 2, 3, 4, 5, 6, 7] in 2023 (especially after the appearance of MIM-based methods) have reported fine-tuning classification, detection, and segmentation results. Therefore, I think requiring the transfer learning results on detection and segmentation is reasonable.
> >
> > Again, we would like to draw the attention of the reviewer that we indeed **have performed a transfer task** following MACL [2], a paper published in ICML, 2023. We wanted to show that our method performs well on transfer tasks of other modalities. The tasks MR, CR, SUBJ, MPQA, SST2, TREC, and MRPC mentioned in Table 5 (Sec. 6.4 of the main manuscript) and Table 11 (Appendix B), are the transfer tasks among all the sentence embedding similarity tasks [3].
> >
> > We again request the reviewer to go through the transfer tasks in our manuscript.
> >
> > > I still think extensive research on temperature in negative-requiring contrastive methods will not bring too much insight into this community.
> >
> > We beg to differ.
> >
> > Recent works like Kukleva et al. (ICLR, 2023) [1], and MACL (ICML, 2023) [2], iSogCLR (ICML, 2023) [4], etc. have explored the role of temperature hyper-parameter in "**negative-requiring**" contrastive learning and found that it plays an important role in the learning process.
> >
> > We have also shown evidence in Table 16 (Appendix I), that the proposed method outperforms several negative-free SSL frameworks on ImageNet100 datasets when trained on the same configurations (same number of epochs and batch size). Hence, like several other researchers, we believe that exploring the role of temperature hyper-parameter is worth considering.
> >
> > We request the reviewer to evaluate the manuscript in that light.
> >
> >
> > ---
> > References
> >
> > [1] Anna Kukleva, Moritz B  ̈ohle, Bernt Schiele, Hilde Kuehne, and Christian Rupprecht. Temperature schedules for self-supervised contrastive methods on long-tail data. In ICLR 2023.
> >
> > [2] Huang Zizheng, Chen Haoxing, Wen Ziqi, Zhang Chao, Li Huaxiong, Wang Bo, and Chen Chunlin. Model-aware contrastive learning: Towards escaping the dilemmas. In ICML, 2023.
> >
> > [3] Tianyu Gao, Xingcheng Yao, and Danqi Chen. SimCSE: Simple contrastive learning of sentence embeddings. In Proceedings of the 2021 Conference on Empirical Methods in Natural Language Processing, pp. 6894–6910
> >
> > [4] Zi-Hao Qiu, Quanqi Hu, Zhuoning Yuan, Denny Zhou, Lijun Zhang, and Tianbao Yang. ,. Not All Semantics are Created Equal: Contrastive Self-supervised Learning with Automatic Temperature Individualization. In ICML 2023

---

### Official Review · Reviewer_7bH3 · 2023-10-31

**Soundness:** 2 fair
**Presentation:** 2 fair
**Contribution:** 2 fair
**Rating:** 3
**Confidence:** 4

**Summary:**

The paper investigate the role of temperature in InfoNCE based SSL methods. The paper also provides analysis to support the construction of the method.

**Strengths:**

- The empirical results looks promising
- The authors provided code to support their method

**Weaknesses:**

- I would advise the authors to continue polishing the presentation. For example, I had a hard time looking at the x-axis and y-axis of Figure 1(d), (e)
- Although terms TP, TN, FN might be well-known, I think it needs to be carefully explained and presented when you are using it under your circumstance. For example, it is unclear to me what is the exact definition of FN pairs in images. How do we measure it? When the representation are not formed, how do we effectively decide what is TP, TN, FN pairs without making mistakes?

**Questions:**

1. For Figure 1(b), why would the ideal convergence only have few points? To me, Figure 1(c) also looks ideal? The paper explains as "the ideal global structure should consist of N closed subser with Minimum intraclass scattering and maximized interclas distance". However, class itself, it a very vague and "human-biased"  term. For example, in imagenet, there are many classes that look very like each other, in this case, would it be ideal to have minimum intraclass scattering? In that case, would it be what's known as the neural collapse in supervised training? So, would that be an "ideal" representation we truly need? I think the motivation here needs to be very carefully explained and justified.

2. In Algo1, s_ij is defined as the cosine sim of the pair (x_i, x_j).  If x_i and x_j are images, do we flatten them and compute the cosine sim? Or do we compute the cosine similarity differently? in the latent space?

3. If we go back to the motivation, would the proposing method giving us better representation? Visually? Is the method learning representation with more intra distance and less inter distance?

4. Does the method incur extra training cost? If so, how does it compare with the gain in performance? Because computing cosine similarity constantly could be a non-trivial increase in terms of computation (resource and time)

---

> ### Author Response · Authors · 2023-11-19
> **Reply to Reviewer 7bH3 (1/2)**
>
> Thanks to the reviewer for acknowledging the strengths of our work
> - empirical results looks promising
> - provided code to support their method
>
> ---
> > [W1] I would advise the authors to continue polishing the presentation. For example, I had a hard time looking at the x-axis and y-axis of Figure 1(d), (e)
>
> Thanks for this comment. We have updated the images for better presentation.
>
> ---
> > [W2] Although terms TP, TN, FN might be well-known, I think it needs to be carefully explained and presented when you are using it under your circumstance. For example, it is unclear to me what is the exact definition of FN pairs in images. How do we measure it? When the representations are not formed, how do we effectively decide what is TP, TN, FN pairs without making mistakes?
>
> We thank the reviewer for this question. We would like to bring the attention of the reviewer to Sec. 3 (below Eqn. 1), where we have already defined True Positive (TP), False Negative (FN) and True Negative (TN) pairs. In the context of contrastive learning, every sample belongs to a class of its own, since we do not have any ground truth label information. In the pre-training stage, we can only say if a pair is positive or negative [2, 3, 4]. However, when dealing with all these pairs, we can argue that there are samples which may belong to the same ground truth class but could be in a negative pair. These pairs are False Negative (FN) pairs. Similarly, there are pairs which contain samples obtained by different augmentations of the same sample, and we term these pairs as True Positive (TP) Pairs. Lastly, pairs containing samples which may belong to different ground truth classes are termed as True Negative (TN) pairs. Only in the evaluation stage, where we can have access to the ground truth label, can we say for sure, which pairs belong to any of these three categories.
>
> ---
> > [Q1] For Figure 1(b), why would the ideal convergence only have few points? To me, Figure 1(c) also looks ideal? The paper explains as "the ideal global structure should consist of N closed subser with Minimum intraclass scattering and maximized interclas distance". However, class itself, it a very vague and "human-biased" term. For example, in imagenet, there are many classes that look very like each other, in this case, would it be ideal to have minimum intraclass scattering? In that case, would it be what's known as the neural collapse in supervised training? So, would that be an "ideal" representation we truly need? I think the motivation here needs to be very carefully explained and justified.
>
> We do agree that the definition of class is subjective. However, many classical algorithms consider that different classes should have higher interclass scattering, whereas samples within a defined class should have minimum intraclass scattering. The term “intraclass scattering” refers to scattering between samples belonging to the same class. On the other hand, the term “Inter-class distance” refers to the distance between the clusters of different classes. Minimum intraclass scattering is possible only if a trained model is able to correctly classify every sample in a class and map every sample of a particular class very close to each other, which is crucial to achieve human-like performance. Also, maximum interclass distance is possible to achieve if the model is able to map samples from different class farther away from each other.
>
> It is obvious that in datasets like ImageNet, and even in small-scale datasets like CIFAR10 or CIFAR100, there are samples from classes which are difficult to differentiate. However, if a model is unable to differentiate between different classes which are very similar to each other, then it can result in a sort of collapse, which is the opposite of the scenario we have mentioned in our motivation. However, the purpose of the illustration was to depict an ideal scenario any learning model aims to achieve after pre-training. Hence, the statement “Minimum intraclass scattering and maximized interclass distance” actually depicts an ideal end state scenario, where the classes are linearly separable and the samples with the same class are mapped close to each other.
>
> However, we want to emphasize that our proposed method is independent of such assumptions and the figures are just given to discuss about feature representation.
>
> ---

---

> ### Author Response · Authors · 2023-11-19
> **Reply to Reviewer 7bH3 (2/2)**
>
> > [Q2] In Algo1, s_ij is defined as the cosine sim of the pair (x_i, x_j). If x_i and x_j are images, do we flatten them and compute the cosine sim? Or do we compute the cosine similarity differently? in the latent space?
>
> After passing the images $x_i$ and $x_j$ through the ResNet encoder and MLP projector, we obtain the feature vectors $z_i$ and $z_j$, respectively. We flatten the vectors to have dimensions $1 \times D$ and L2-normalize the vectors $z_i$ and $z_j$. To calculate the cosine similarity, we simply take the dot product of $z_i$ and $z_j$. So, as the reviewer has correctly pointed out, the cosine similarity is calculated in the latent space.
>
> ---
>
> > [Q3] If we go back to the motivation, would the proposing method giving us better representation? Visually? Is the method learning representation with more intra distance and less inter distance?
>
> From the tables presented below, we can take a look at the overall uniformity and inter-class uniformity values for SimCLR, MoCov2, DCLW, DCL, and DySTreSS. Overall uniformity takes all the samples in the feature space and is calculated as defined in [1]. We define another term, Inter-class uniformity which is defined by Eqn. 19 in Appendix E.
>
> Table 1. Overall Uniformity and Inter-Class Uniformity values for SimCLR, MoCov2, DCL, DCLW, DySTreSS for CIFAR10 dataset.
> | CIFAR10 | | | |
> |----------|-----------|-----------------|----------|
> | Method | Unifomity | InterClass Unif | Accuracy |
> | SimCLR [3] | -2.2135 | -0.5194 | 83.65 |
> | MoCov2 [5]| -2.1929 | -0.5094 | 83.67 |
> | DCLW [6]| -2.5482 | -0.5349 | 84.02 |
> | DCL [6]| -2.7717 | -0.5524 | 84.47 |
> | DySTreSS | -2.8430 | -0.5797 | 85.83 |
>
> Table 2. Overall Uniformity and Inter-Class Uniformity values for SimCLR, MoCov2, DCL, DCLW, DySTreSS for CIFAR100 dataset.
> | CIFAR100 | | | |
> |----------|-----------|-----------------|----------|
> | Method | Unifomity | InterClass Unif | Accuracy |
> | SimCLR [3]| -2.2556 | -0.5246 | 52.32 |
> | MoCov2 [5]| -2.2202 | -0.5364 | 54.01 |
> | DCLW [6]| -2.6865 | -0.6405 | 55.87 |
> | DCL [6]| -2.8314 | -0.6792 | 56.02 |
> | DySTreSS | -2.8169 | -0.7657 | 56.72 |
>
> Table 3. Overall Uniformity and Inter-Class Uniformity values for SimCLR, MoCov2, DCL, DCLW, DySTreSS for ImageNet1K dataset.
> | ImageNet1K | | | |
> |----------|-----------|-----------------|----------|
> | Method | Unifomity | InterClass Unif | Accuracy |
> | SimCLR [3]| -2.9084 | -0.8739 | 63.2 |
> | DCLW [6]| -3.0137 | -0.9544 | 64.2 |
> | DCL [6]| -3.0786 | -1.0053 | 65.1 |
> | DySTreSS | -3.0597 | -1.0015 | 65.2 |
>
> We can see that the uniformity and the inter-class uniformity values decreases, which indicates that the inter-cluster distance increases from SimCLR to DySTreSS, evident from the increasing accuracy values, as well. Lower inter-cluster uniformity means better separability between clusters. Hence, we can argue that the proposed model learns more separable classes, that is, more inter-class distance.
>
> ---
>
> > [Q4] Does the method incur extra training cost? If so, how does it compare with the gain in performance? Because computing cosine similarity constantly could be a non-trivial increase in terms of computation (resource and time)
>
> No, the proposed method does not incur extra training costs for computing cosine similarity constantly, when compared to existing methods like SimCLR. It is computing the same cosine similarity matrix as done in many other contrastive SSL methods, e.g. SimCLR, MoCo, DCL.
>
> ---
>
> References:
>
> [1] Tongzhou Wang and Phillip Isola. Understanding contrastive representation learning through alignment and uniformity on the hypersphere. In ICML, 2020.
>
> [2] I. Misra and L. van der Maaten, "Self-Supervised Learning of Pretext-Invariant Representations," in 2020 IEEE/CVF Conference on Computer Vision and Pattern Recognition
>
> [3] Ting Chen, Simon Kornblith, Mohammad Norouzi, and Geoffrey Hinton. 2020. A simple framework for contrastive learning of visual representations. In Proceedings of the 37th International Conference on Machine Learning (ICML'20)
>
> [4] He, K., Fan, H., Wu, Y., Xie, S., & Girshick, R.B. (2019). Momentum Contrast for Unsupervised Visual Representation Learning. 2020 IEEE/CVF Conference on Computer Vision and Pattern Recognition (CVPR), 9726-9735.
>
> [5] Chen, X., Fan, H., Girshick, R.B., & He, K. (2020). Improved Baselines with Momentum Contrastive Learning. ArXiv, abs/2003.04297.
>
> [6] Yeh, CH., Hong, CY., Hsu, YC., Liu, TL., Chen, Y., LeCun, Y. (2022). Decoupled Contrastive Learning. In: Avidan, S., Brostow, G., Cissé, M., Farinella, G.M., Hassner, T. (eds) Computer Vision – ECCV 2022.

---

> ### Comment · Reviewer_7bH3 · 2023-11-22
>
> I thank the authors for clarification and rebuttal. Here are some additional comments based on the rebuttal:
> - Presentation: I appreciate the authors taking the efforts to improve the presentation. I would recommend the authors continue to polish areas such as typos. For example, the rebuttal starts with "acknowledging". However, this point is minor compared to the rest.
> - Performance of baseline methods: I understand from the paper that methods pretrain with 200 epochs. This approach is ok if the authors tries to claim method's efficiency. However, it is not convincing if the authors use 200 epoch result and try to claim "We can see that the uniformity and the inter-class uniformity values decreases, which indicates that the inter-cluster distance increases from SimCLR to DySTreSS, evident from the increasing accuracy values, as well. Lower inter-cluster uniformity means better separability between clusters. Hence, we can argue that the proposed model learns more separable classes, that is, more inter-class distance." in the rebuttal. Since, it could be very likely that the methods take longer to converge. For example, it is known that swav and byol seem to converge faster which could also end up with a higher accuracy and better inter-intra similarity
> - Definition on TP, TN,FN: I apologize for not clearing my words. Yes, I am aware that the authors defined these terms below equation (1). This circles back to presentation issue. For such important terms, I would recommend either i. formally define the term ii. At least put a (TP), (TN), (FN) next to where you define it to help the reader better understand
> - [Q1] For Figure 1(b), why would the ideal convergence only have few points? To me, Figure 1(c) also looks ideal?: I thank the authors for providing explanation. However, I find the explanation very unconvincing because
> (1) If as the author states: "However, we want to emphasize that our proposed method is independent of such assumptions and the figures are just given to discuss about feature representation", it is somewhat troubling to see something irrelevant with the assumption and motivation appear in the work.
> (2) "It is obvious ... to each other.": I would argue that in more complicated real-world datasets such as ImageNet-1k, classes are "correlated". Like many works have pointed out that they can even be decomposed into a superposition of concepts. The authors suggest "However, if a model is unable to differentiate between different classes which are very similar to each other, then it can result in a sort of collapse, which is the opposite of the scenario we have mentioned in our motivation. ", I don't understand what do you mean by "sort of collapse". If say, even in the CIFAR-10 case, truck and car are completely separated, is this a desired representation? I think making such claim can be dangerous and I would strongly recommend the authors to think more carefully and polish the writing and terms to avoid conveying any unjustified message.
>
>
> As a result of the reasons and questions above, I would retain my score.

---

> > ### Author Response · Authors · 2023-11-22
> > **Reply to Reviewer 7bH3**
> >
> > > Presentation: I appreciate the authors taking the efforts to improve the presentation. I would recommend the authors continue to polish areas such as typos. For example, the rebuttal starts with "acknowledging". However, this point is minor compared to the rest.
> >
> > Sincerely sorry for the typo.
> >
> > > Performance of baseline methods: I understand from the paper that methods pretrain with 200 epochs. . . . Hence, we can argue that the proposed model learns more separable classes, that is, more inter-class distance." in the rebuttal. Since, it could be very likely that the methods take longer to converge. For example, it is known that swav and byol seem to converge faster which could also end up with a higher accuracy and better inter-intra similarity
> >
> > We agree that there are models that may converge faster. However, we can say that when pre-trained for **200 epochs**, our proposed model performs the best. We would also like to thank the reviewer for correctly pointing out the issue. As per the previous question of the reviewer, the motivation was to find out a relation between the performance (Top-1 accuracy), and the feature representation. For some other value of the epoch hyper-parameter, some other algorithm might have a better feature representation. We have clearly mentioned that in Appendix E of the revised manuscript.
> >
> > ---
> > > Definition on TP, TN,FN: I apologize for not clearing my words. Yes, I am aware that the authors defined these terms below equation (1). This circles back to presentation issue. For such important terms, I would recommend either i. formally define the term ii. At least put a (TP), (TN), (FN) next to where you define it to help the reader better understand
> >
> > Thanks to the reviewer for this suggestion. We have made the necessary changes as per the suggestions in Sec. 3.
> >
> > ---
> > > (1) If as the author states: "However, we want to emphasize that our proposed method is independent of such assumptions and the figures are just given to discuss about feature representation", it is somewhat troubling to see something irrelevant with the assumption and motivation appear in the work.
> >
> > After closely reviewing the comment, we have decided to remove Fig. 1(a)-(c) to avoid any confusion. We want to reiterate that those figures were not related to our methodology. Thus, the modification does not affect the reported results.
> >
> > > (2) "It is obvious ... to each other.": I would argue that in more complicated real-world datasets such as ImageNet-1k, classes are "correlated". Like many works have pointed out that they can even be decomposed into a superposition of concepts. The authors suggest "However, if a model is unable to differentiate between different classes which are very similar to each other, then it can result in a sort of collapse, which is the opposite of the scenario we have mentioned in our motivation. ", I don't understand what do you mean by "sort of collapse". If say, even in the CIFAR-10 case, truck and car are completely separated, is this a desired representation? I think making such claim can be dangerous and I would strongly recommend the authors to think more carefully and polish the writing and terms to avoid conveying any unjustified message.
> >
> > As mentioned in the previous comment, we agree with the reviewer that the notion of class is subjective, and the desired representation may or may not be completely separated. To avoid confusion, we have removed the representative diagrams and the related discussion. We would like to thank the reviewer for the insightful comment.

---

### Official Review · Reviewer_S1Ew · 2023-11-01

**Soundness:** 2 fair
**Presentation:** 2 fair
**Contribution:** 2 fair
**Rating:** 5
**Confidence:** 4

**Summary:**

The authors focus on improving the performance of InfoNCE loss by proposing a cosine-similarity dependent temperature scaling function. The authors also provide experimental results to demonstrate the effectiveness of the proposed method.

**Strengths:**

1. The writing of this paper is clear, and the descriptions and justifications of the methods are comprehensible.
2. This paper provides a comprehensive analysis of the impact of temperature coefficients on feature representation in contrastive learning.

**Weaknesses:**

1. The design of Algorithm 1 in the paper is merely based on certain rules and lacks theoretical underpinnings.
2. While the paper mentions that adjusting the temperature coefficient can improve feature distribution, corresponding results are not presented in the experimental section.

**Questions:**

1. On what basis is the cosine function used in Algorithm 1? Are there any theoretical results to support this choice?
2. Is there experimental evidence to support the claim that adjusting the temperature parameter can improve feature representation?
3. Why wasn't a comparison made with the method proposed by Kukleva et al. [1]?
4. There was a recent paper [2] in ICML that utilized individualized temperature parameters to optimize the contrastive learning loss. How do these two papers differ?



[1] Kukleva et al,. Temperature schedules for self-supervised contrastive methods on long-tail data. In ICLR 2023.
[2] Qiu et al,. Not All Semantics are Created Equal: Contrastive Self-supervised Learning with Automatic Temperature Individualization. In ICML 2023.

---

> ### Author Response · Authors · 2023-11-19
> **Response to Reviewer S1Ew (1/3)**
>
> Thank you for acknowledging the strengths of our work:
> - the descriptions and justifications of the methods are comprehensible.
> - provides a comprehensive analysis of the impact of temperature coefficients on feature representation in contrastive learning.
> ---
> > [W1] The design of Algorithm 1 in the paper is merely based on certain rules and lacks theoretical underpinnings.
>
> Thanks for your comment. Please note that our paper is not merely based on certain rules. The criteria (1)-(4) as mentioned in Sec. 4.3 arises from the intuitive explanations of the effect of temperature on the local and global structures as mentioned in Sec. 4.2. However, the temperature function in Algorithm 1 is not based on those criteria, but it is a result of the derived temperature function that comes from Equation 14 in Appendix C. As can be seen in Sec. 4.3, the equations 6 and 7 arise as a natural consequence of differentiating the InfoNCE loss with respect to the cosine similarity of the positive pair $s_{ii+}$ and negative pairs $s_{ij}$. Considering that the temperature hyper-parameter $\tau$ is a function of cosine similarity. We take no other assumption in our work. In Proposition 1, we prove that the nature of the slope arises as a consequence of solving the ODE resulting from Equations 6 and 7. Only after that, we propose a temperature function in Algorithm 1. In Appendix C, we have derived the mathematical formulation of the temperature function, and provided necessary insights from the equations as mentioned in Equation 14 and 15 of Appendix C.  This  shows that our proposition is not based on certain rules and has a theoretical background.
>
> ---
> > [W2] While the paper mentions that adjusting the temperature coefficient can improve feature distribution, corresponding results are not presented in the experimental section.
>
> Thanks to the reviewer for pointing this out. As a metric of the feature distribution, we present the overall uniformity and inter-class uniformity values for SimCLR, DCL, DCLW, MoCov2 and DySTreSS (proposed) on CIFAR10, CIFAR100 and ImageNet1K datasets. Uniformity values denote how well the samples are distributed over the feature space [3]. We have included the tables for the same in the reply to your Q2, and in Table 11 of Appendix E of the manuscript as well. In all the datasets, our proposed method shows lowest overall uniformity and inter-class uniformity, which indicate that the proposed method learns more separable clusters than the SOTA methods.
>
> ---
> > [Q1] On what basis is the cosine function used in Algorithm 1? Are there any theoretical results to support this choice?
>
> We thank the reviewer for asking this insightful question. Please note that the cosine function used in Algorithm 1 has theoretical support and is discussed as follows. We would like to draw the attention of the reviewer to the fact that we have shown a derivation of the nature of the slope of the temperature function in the negative and positive half of the cosine-similarity vs. temperature plane as shown in Fig. 8 of Appendix C. The derivation starts with the basic postulate that the term $\frac{\partial \mathcal{L}}{\partial s_{ij}}$ should be positive when $s_{ij}$ is the cosine similarity of the samples in a negative pair. Whereas, for the positive pairs, the term $\frac{\partial \mathcal{L}}{\partial s_{ij}}$ should be negative as with the decrease in the loss, the cosine similarity between the samples in a positive pair increases. By solving the resulting differential equation, we get an expression for the temperature function given by Eqn. 14. Plotting the function in Fig. 8, we get an idea about the slopes of the temperature function. Any function following these criteria can be used as a temperature function.
>
> We have also added now results by using different temperature functions (cosine, linear and exponential), in the Appendix G of the revised manuscript which show that using any function which satisfy the conditions in Proposition 1 (and subsequently proved in Appendix C), that is, positive and negative slope in the positive and negative half of the temperature vs. cosine-similarity plane, yields satisfactory performance. As cosine function follows this proposition and is differentiable everywhere, we have selected it for all our experiments.

---

> ### Author Response · Authors · 2023-11-19
> **Response to Reviewer S1Ew (2/3)**
>
> > [Q2] Is there experimental evidence to support the claim that adjusting the temperature parameter can improve feature representation?
>
> Thanks for the comment. We would like to mention that there is experimental evidence and it is discussed as follows. In the tables given below, we have provided overall uniformity and inter-class uniformity values for 5 contrastive learning algorithms on CIFAR10, CIFAR100 and ImageNet1K datasets to support the claim.
>
> Table 1. Overall Uniformity and Inter-Class Uniformity values for SimCLR, MoCov2, DCL, DCLW, DySTreSS for CIFAR10 dataset.
> | CIFAR10 | | | |
> |----------|-----------|-----------------|----------|
> | Method | Unifomity | InterClass Unif | Accuracy |
> | SimCLR | -2.2135 | -0.5194 | 83.65 |
> | MoCov2 | -2.1929 | -0.5094 | 83.67 |
> | DCLW | -2.5482 | -0.5349 | 84.02 |
> | DCL | -2.7717 | -0.5524 | 84.47 |
> | DySTreSS | -2.8430 | -0.5797 | 85.83 |
>
> Table 2. Overall Uniformity and Inter-Class Uniformity values for SimCLR, MoCov2, DCL, DCLW, DySTreSS for CIFAR100 dataset.
> | CIFAR100 | | | |
> |----------|-----------|-----------------|----------|
> | Method | Unifomity | InterClass Unif | Accuracy |
> | SimCLR | -2.2556 | -0.5246 | 52.32 |
> | MoCov2 | -2.2202 | -0.5364 | 54.01 |
> | DCLW | -2.6865 | -0.6405 | 55.87 |
> | DCL | -2.8314 | -0.6792 | 56.02 |
> | DySTreSS | -2.8169 | -0.7657 | 56.72 |
>
> Table 3. Overall Uniformity and Inter-Class Uniformity values for SimCLR, MoCov2, DCL, DCLW, DySTreSS for ImageNet1K dataset.
> | ImageNet1K | | | |
> |----------|-----------|-----------------|----------|
> | Method | Unifomity | InterClass Unif | Accuracy |
> | SimCLR | -2.9084 | -0.8739 | 63.2 |
> | DCLW | -3.0137 | -0.9544 | 64.2 |
> | DCL | -3.0786 | -1.0053 | 65.1 |
> | DySTreSS | -3.0597 | -1.0015 | 65.2 |
>
> From the above tables, we can see 2 quantities, overall uniformity and inter-class uniformity. Overall uniformity takes all the samples in the feature space and is calculated as defined in [1]. We define another term, Inter-class uniformity which is defined by Eqn. 19 in Appendix E.
>
> A lower uniformity value indicates that the samples are more spread out than a model with a higher uniformity value. The inter-class uniformity metric value indicates how farther apart the class centroids are from one another. Hence, a lower inter-class uniformity value indicates that the clusters are more separated than a mode with a higher value. We see that the overall uniformity values are the lowest for our proposed method. In fact, we can see a trend in the uniformity values with the accuracy for different models as well. In our case, the inter-class uniformity metric is the lowest, which means that the class centroids are farther away than the other models, and is an indicator of how easily separable the clusters are for our proposed method. Thus, we can say that the proposed framework improves feature representations, and hence, learns more separable clusters, resulting in higher 200-NN accuracy.
>
> ---
> > [Q3] Why wasn't a comparison made with the method proposed by Kukleva et al. [1]?
>
> Thanks for your suggestion and based on your suggestion we have made this comparison now and it is discussed as follows. To check the performance of our proposed framework in comparison to Kukleva et al. [1], where the authors trained their models for 2000 epochs on Long-Tailed datasets. For fair comparison we also trained our proposed framework for the same number of epochs  (2000 epochs) on CIFAR10-LT, CIFAR100-LT. In the table below we report the 10-NN accuracy (as per Kukleva et al.) of both the datasets. For the experiments on ImageNet100-Lt, we used a batch size of 256 on a 24GB NVIDIA A5000 GPU. From the results it can be seen that our proposed framework outperforms Kukleva et al. on long tailed datasets.
>
> Table 4. Comaprison of performance on benchmark long-tailed datasets with Kukleva et al. [1]
> |  | CIFAR10-LT | CIFAR100-LT | ImageNet100-LT |
> |---|---|---|---|
> | Kukleva et al. [1] | 62.91 | 30.20 | 45.3 (repro.) |
> | DySTreSS | 64.98 | 31.71 | 46.1 |
>
> We have added the above comparison in our revised manuscript.
>
> ---

---

> ### Author Response · Authors · 2023-11-19
> **Response to Reviewer S1Ew (3/3)**
>
> > [Q4] There was a recent paper [2] in ICML that utilized individualized temperature parameters to optimize the contrastive learning loss. How do these two papers differ?
>
>  In this ICML paper [2], the authors attempt to implement distributionally robust optimization (DRO) on an existing algorithm SogCLR, which leads to temperature individualization. However, it requires the use of a temperature hyper-parameter $\tau_i$ corresponding to each anchor sample, and it is updated at each iteration. In our opinion, this no longer keeps the temperature as a hyper-parameter and converts it into another optimizable parameter. The authors also claim that by doing so, the global semantics is taken into account during pre-training.
>
> However, in our case, the temperature hyper-parameter is not dependent on each anchor sample. Rather, we devise a temperature function which gives a suitable non-learnable temperature for each pair of samples in contrastive learning.
>
>
> Thus, the two papers differ significantly.
>
> We have included the above discussion in the “Related Works” section of the revised manuscript.
>
> To compare our method with this ICML paper [2], we ran our proposed method for 400 epochs, following the configuration in [2], and the results are reported below.
>
> Table 5. Comparion with iSogCLR [2] on CIFAR datasets
> || CIFAR10 | CIFAR100 |
> |---|---|---|
> |iSogCLR [2] | 89.24 | 63.82 |
> |DySTreSS | 90.34 | 63.94 |
>
> ---
> References:
>
> [1] Kukleva et al,. Temperature schedules for self-supervised contrastive methods on long-tail data. In ICLR 2023.
>
> [2] Qiu et al,. Not All Semantics are Created Equal: Contrastive Self-supervised Learning with Automatic Temperature Individualization. In ICML 2023.
>
> [3] Tongzhou Wang and Phillip Isola. Understanding contrastive representation learning through alignment and uniformity on the hypersphere. In ICML, 2020.

---

### Author Response · Authors · 2023-11-19
**Rebuttal (Part 1)**

We thank the reviewers for their invaluable feedback. In this comment, we give a summary of the changes made to the manuscript.

In the individual comments, we have addressed the comments and questions from the reviewers.

---
## Summary of Changes
---

### Main Text
---
- [**S1Ew**] (Section 6.3) Added comparison of the proposed framework with Kukleva et al. [1] on small-scale benchmarks.
- [**7bH3**] (Section 4.2) Improved Fig.1 for better presentation.
- [**7bH3**, **RTHa**] (Section 3) An improved sentence for defining True Positive, False Negative, and True Negative pairs.
- [**RTHa**] (Section 4.2, 4.3) Changes made to some notation for better understanding
- [**iD3A**] (Section 2) Added discussion of negative-free frameworks in "Related Works"

---
### Appendix
---

- [**S1Ew**] (Appendix A.2) Added comparison of the proposed framework with Kukleva et al. [1] on CIFAR and ImageNet1K datasets.
- [**S1Ew**, **7bH3**] (Appendix E) Added Table 12 containing the overall uniformity, and inter-class uniformity values for self-supervised contrastive learning algorithms for CIFAR10, CIFAR100, and ImageNet1K datasets.
- [**iD3A**] (Appendix I) Added comparison with non-contrastive frameworks DINO [2] and WMSE [3] on CIFAR datasets.
- [**RTHa**] (Appendix G) Added comparison of proposed cosine temperature function with other temperature functions.
- [**RTHa**] (Appendix H) Added comparison of proposed cosine temperature function with monotonic temperature functions

---
References:

[1] Kukleva et al,. Temperature schedules for self-supervised contrastive methods on long-tail data. In ICLR 2023.

[2] Caron M, Touvron H, Misra I, et al. Emerging properties in self-supervised vision transformers. Proceedings of the IEEE/CVF international conference on computer vision. 2021: 9650-9660.

[3] Ermolov A, Siarohin A, Sangineto E, et al. Whitening for self-supervised representation learning. International Conference on Machine Learning. PMLR, 2021: 3015-3024.

---

> ### Author Response · Authors · 2023-11-21
> **Rebuttal (Part 2)**
>
> ## Summary of Changes
> ---
> ### Appendix
> ---
> - [**iD3A**] (Appendix I) Added comparison with DINO [2], WMSE [3], Zero-CL [4], and ARB [5] on the ImageNet100 dataset.
>
> ---
> References (Contd. from Part 1):
>
> [4] Zhang S, Zhu F, Yan J, et al. Zero-cl: Instance and feature decorrelation for negative-free symmetric contrastive learning[C]//International Conference on Learning Representations. 2021.
>
> [5] [4] Zhang S, Qiu L, Zhu F, et al. Align representations with base: A new approach to self-supervised learning. Proceedings of the IEEE/CVF Conference on Computer Vision and Pattern Recognition. 2022: 16600-16609.

---

> > ### Author Response · Authors · 2023-11-22
> > **Rebuttal (Part 3)**
> >
> > ## Summary of Changes
> > ---
> > ### Main Text
> > ---
> > - [**7bH3**] (Section 4) Changed Figure 1
> >
> > ---
> > ### Appendix
> > ---
> > - [**iD3A**] (Appendix I) Updated Table 16. Added results for Barlow Twins [6] and SimSiam [7].
> >
> > ---
> > References (Contd. from Part 2)
> >
> > [6] Jure Zbontar, Li Jing, Ishan Misra, Yann LeCun, and St  ́ephane Deny. Barlow twins: Self-supervised learning via redundancy reduction. In International Conference on Machine Learning, pp. 12310–12320. PMLR, 2021
> >
> > [7] Xinlei Chen and Kaiming He. Exploring simple siamese representation learning. In CVPR, 2021.

---

### Meta-Review · Area_Chair_SbNt · 2023-12-04

**Metareview:**

The paper proposes an adaptive temperature scaling for contrastive learning. The main concern raised by the reviewers is the intuition and theoretical justification of the proposed temperature scaling (for instance, the use of $\cos$ function). There are other concerns about the clarity of the definitions and the weakness/insufficiency of the experimental evidence.

**Justification For Why Not Higher Score:**

The main concern is with the theoretical intuition/justification of the proposed temperature scaling. There are other concerns about the clarity of the definitions and the weakness/insufficiency of the experimental evidence. Despite one of the reviewers increasing their score, the overall feedback by the reviewers is not positive enough for acceptance.

**Justification For Why Not Lower Score:**

N/A

---

### Decision · Program_Chairs · 2024-01-16

Reject